# Warming enhances carbon dioxide and methane fluxes from Red Sea seagrass (*Halophila stipulacea*) sediments

Celina Burkholz[1,2], Neus Garcias-Bonet[1], and Carlos Manuel Duarte[1]

[1] Red Sea Research Center (RSRC) and Computational Bioscience Research Center (CBRC), King Abdullah University of Science and Technology (KAUST), Thuwal, Saudi Arabia

[2] UWA Oceans Institute and School of Biological Sciences, The University of Western Australia, Crawley, WA, Australia

*Correspondence to:* Carlos M. Duarte (carlos.duarte@kaust.edu.sa)

**Abstract.** Seagrass meadows are autotrophic ecosystems acting as carbon sinks, but they have also been shown to be sources of carbon dioxide ($CO_2$) and methane ($CH_4$). Seagrasses can be negatively affected by increasing seawater temperatures, but the effects of warming on $CO_2$ and $CH_4$ fluxes in seagrass meadows have not yet been reported. Here, we examine the effect of two disturbances on air-seawater fluxes of $CO_2$ and $CH_4$ in Red Sea *Halophila stipulacea* communities compared to adjacent unvegetated sediments using cavity ring-down spectroscopy. We first characterized $CO_2$ and $CH_4$ fluxes in vegetated and adjacent unvegetated sediments, and then experimentally examined their response, along with that of the C isotopic signature of $CO_2$ and $CH_4$, to gradual warming from 25 ˚C (winter seawater temperature) to 37 ˚C, 2 ˚C above current maximum temperature. In addition, we assessed the response to prolonged darkness, thereby providing insights into the possible role of suppressing plant photosynthesis in supporting $CO_2$ and $CH_4$ fluxes. We detected 6-fold higher $CO_2$ fluxes in vegetated compared to bare sediments, as well as 10- to 100-fold higher $CH_4$ fluxes. Warming led to an increase in net $CO_2$ and $CH_4$ fluxes, reaching average fluxes of $10,422.18 \pm 2,570.12$ μmol $CO_2$ m$^{-2}$ d$^{-1}$ and $88.11 \pm 15.19$ μmol $CH_4$ m$^{-2}$ d$^{-1}$, while $CO_2$ and $CH_4$ fluxes decreased over time in sediments maintained at 25 ˚C. Prolonged darkness led to an increase in $CO_2$ fluxes but a decrease in $CH_4$ fluxes in vegetated sediments. These results add to previous research identifying Red Sea seagrass meadows as a significant source of $CH_4$, while also indicating that sublethal warming may lead to increased emissions of greenhouse gases from seagrass meadows, providing a feedback mechanism that may contribute to further enhance global warming.

## 1 Introduction

Global warming, as a result of anthropogenic emissions of greenhouse gases, has led to ocean warming by 0.11 ˚C between 1971 to 2010 (IPCC, 2014) with the global mean sea-surface temperature predicted to increase further with additional emissions, depending on emission scenarios (IPCC, 2014). Ocean warming is leading to a shift in species and ecosystem processes (Hoegh-Guldberg and Bruno, 2010), including metabolic processes that are under strong thermal control (Brown et al., 2004; Garcias-Bonet et al., 2018, 2019b).

Ecosystem metabolism can also be a source of greenhouse gases, depending on the metabolic balance of the community, where autotrophic communities [net community production (NCP) > 0] act as a sink for carbon dioxide ($CO_2$), while heterotrophic communities [net community production (NCP) < 0] act as a source of $CO_2$ (Duarte et al., 2011). Since respiration rates tend to increase faster with warming than primary production does (Brown et al., 2004; Harris et al., 2006; Regaudie-De-Gioux and Duarte, 2012), warming may lead to typically autotrophic ecosystems, such as seagrass meadows, shifting to net heterotrophic, thereby switching from acting as sinks to sources of $CO_2$ (Harris et al., 2006). Emissions of metabolic greenhouse gases with ocean warming may provide a feedback mechanism by which anthropogenic emissions of greenhouse gases may lead to warming-dependent emissions by coastal ecosystems, therefore enhancing climate warming. This feedback effect is particularly likely to occur where methane ($CH_4$) is released, as $CH_4$ is calculated to have a global warming potential 28 times larger than $CO_2$ per mol of carbon C emitted (Myhre et al., 2013).

Indeed, $CO_2$ and $CH_4$ emissions from some tropical mangrove forests have been calculated to partially offset the capacity of mangroves to act as C sinks (Rosentreter et al., 2018). Whereas the emission of $CO_2$ and $CH_4$ from seagrass ecosystems has received far less attention, seagrass ecosystems have been reported to support $CH_4$ emissions of the order of 1.4 to 401.3 $\mu$mol $CH_4$ $m^{-2}$ $d^{-1}$ (cf. Table 1 in Garcias-Bonet and Duarte (2017)). Provided estimates of their global extent of seagrass meadows ranging from a documented 326,000 $km^2$ (Unsworth et al., 2018) to a predicted 1.6 million $km^2$ (Jayathilake and Costello, 2018), seagrass meadows may be important, yet hitherto overlooked contributors to $CH_4$ emissions. Garcias-Bonet and Duarte (2017) reported that seagrasses could contribute to global $CH_4$ emissions by releasing $CH_4$ at a rate of 0.09 - 2.7 Tg $yr^{-1}$, which may increase the contribution of marine global emissions to previously reported global estimates by about 30 % (Garcias-Bonet and Duarte, 2017).

Seagrasses are known to be autotrophic ecosystems, acting as C sinks (Duarte et al., 2010) supporting a global burial rate of 27.4 Tg C $yr^{-1}$ (Duarte et al., 2005). They store carbon in their below- and above-ground biomass on a short term, as well as in their sediment on a long-term (Duarte et al., 2005). They account for 10 % of the C storage in ocean sediments even though they only cover 0.2 % of the ocean surface (Duarte et al., 2005; Fourqurean et al., 2012). However, disturbances can lead to the loss of biomass and the emissions of stored C turning blue carbon ecosystems into C sources (Macreadie et al., 2015; Lovelock et al., 2017; Arias-Ortiz et al., 2018) which will ultimately contribute to global emissions intensifying the greenhouse effect. Lyimo et al. (2018) showed that stressors such as shading and grazing led to an increase of $CH_4$ emissions by seagrass ecosystems by reducing their photosynthetic capacity. Garcias-Bonet and Duarte (2017) reported that $CH_4$ release from seagrass sediments tended to increase with seawater temperature, and suggested that $CH_4$ emissions by seagrass ecosystems may be under temperature control in the Red Sea. Indeed, some seagrass ecosystems in the Red Sea have shown to shift from an autotrophic to a heterotrophic state during the warmer summer months, indicating that some seagrass communities might already grow past their thermal optimum (Burkholz et al., 2019a).

The Red Sea ranks as the warmest sea in the world, with summer seawater temperatures reaching 35 ˚C, and is warming at higher rates (0.17 ± 0.07 °C $decade^{-1}$, Chaidez et al., 2017) than those of the global ocean (0.11 °C $decade^{-1}$, Rhein et al., 2013). Provided respiration rates and also $CH_4$ fluxes in seagrass ecosystems are likely to increase with temperature, seagrass meadows in the Red Sea may be close to shifting from net sinks to net sources of greenhouse gases with further warming. Emission rates are also dependent on organic carbon supply, as high sediment organic matter can promote an

increase in $CH_4$ production (Sotomayor et al., 1994; Gonsalves et al., 2011) and organic matter released from seagrass photosynthesis may also stimulate $CO_2$ and $CH_4$ production in the sediment community. Indeed, sediments in seagrass ecosystems support a 1.7-fold higher organic matter content than surrounding bare sediments, not only due to the slow turn-over of biomass but also due to their ability to trap particles (Kennedy et al., 2010; Duarte et al., 2013).

Here, we test the hypothesis that $CO_2$ and $CH_4$ fluxes by seagrass communities increase with warming. We do so by experimentally examine the effect of warming and plant activity on air-seawater fluxes of $CO_2$ and $CH_4$ in a Red Sea seagrass (*Halophila stipulacea)* community. The tropical seagrass species *Halophila stipulacea* (Forsskål) Ascherson is native to the Indian Ocean and is one of the most common species in the Red Sea (Qurban et al., 2019). It seems to be highly adaptive to various environments, as it is now found as an exotic species in the Mediterranean (Lipkin, 1975) and the

Caribbean Sea (Ruiz and Ballantine, 2004), indicating its high resilience to changing conditions (Por, 1971). We first characterize air-seawater fluxes of $CO_2$ and $CH_4$ in Red Sea *Halophila stipulacea* communities compared to adjacent unvegetated sediments, and then experimentally examine their response, along with that of the C isotopic signature of $CO_2$ and $CH_4$, to gradual warming from 25 ˚C to 37 ˚C. In addition, we assess the response to prolonged darkness, thereby providing insights into the possible role of plant photosynthesis in supporting $CO_2$ and $CH_4$ fluxes.

## 2 Material and methods

### 2.1 Study site and sample collection

Samples were collected at Al Kharar, a lagoon on the Saudi coast of the central Red Sea in February 2018. Two *H. stipulacea* meadows at a depth of 2-3 m, S1 (22˚56'46.5"N, 38˚52'40.6"E) and S2 (22˚54'44.5"N, 38˚53'50.9"E), were

chosen to represent a range of organic matter content in the sediment, selected to evaluate greenhouse gas fluxes. Moreover, the *H. stipulacea* meadow in the middle of the lagoon (S2) with higher biomass density (Table 1) was chosen as the study site to experimentally assess the role of temperature and darkness in greenhouse gas fluxes. The seagrass and sediment community was sampled using translucent cylindrical PVC cores (26 cm length and 9.5 cm in diameter). The sharpened edge of the core was carefully pushed approximately 10 cm into the sediment with a rubber hammer so that the structure of

leaves, roots and sediment stayed intact. A rubber stopper was then placed on top, before the core was carefully pulled out of the sediment without disturbing the structure and another rubber stopper was placed on the bottom of the core. The sediment cores were immediately transported to the laboratory.

### 2.2 Sediment and plant characterization

Once the cores were opened, the first 10 cm of the sediment and the plant biomass from the same cores were collected and dried at 60 ˚C to a constant dry weight. To characterize the two different *H. stipulacea* meadows, sediment and plant biomass samples were then ground to analyze the sediment composition and conduct nutrient analyses. A 50 mL tube was filled with sediment from the first 10 cm and the contents dried at 60 ˚C to a constant dry weight and weighed to determine the sediment bulk density (g sediment $cm^{-3}$). Organic matter content was analyzed by loss on ignition (LOI, Dean, 1974).

Approx. 5 g of dried sediment were placed in a muffle furnace and burned at 450 ˚C for 5 hours. The organic matter content was calculated as:

$$\%OM = \frac{pre-ignition\ weight(g) - post-ignition\ weight\ (g)}{pre-ignition\ weight\ (g)} \times 100 \qquad (1)$$

The carbonate content was estimated using a Pressure Gauge Calcimeter. Approx. 1 g of sample was placed in the calcimeter and the recipient was filled with 10 % hydrochloric acid (HCl). The mass of $CaCO_3$ in the sample (g) was then calculated as

follows:

$$m_{CaCO_3} = \frac{p - b}{a \times w} \qquad (2)$$

where $p$ is the pressure recorded (ppm), $b$ is the slope and $a$ the intercept derived from the calibration curve, and $w$ is the exact weight of each sediment sample (g). The percentage of $CaCO_3$ in the sample (% DW) was then calculated using Eq. 3:

$$\%_{CaCO_3} = \frac{m_{CaCO_3}}{w \times 100} \qquad (3)$$

Dried sediment and plant samples were digested using USEPA method 3052 and analyzed with nitric acid ($HNO_3$) and HCl using USEPA method 200.7 following manufacturer instructions. The phosphorus content (% DW) was analyzed using inductively coupled plasma optical emission spectroscropy (ICP-OES) on an Agilent 5110 ICP-OES (Agilent Technologies, Santa Clara, CA, USA). The C and N concentration of both plants and sediments was analyzed after acidification with HCl (Hedges and Stern, 1984), using Flash 2000 Organic Elemental Analyzer (CHNS/O-2, Thermo Fisher Scientic, Waltham,

MA, USA). The isotopic signature of $^{13}C$ in sediment organic matter was analyzed, using cavity ring-down spectroscropy (CRDS G2201-I, Picarro Inc., Santa Clara, CA, USA), from the $^{13}C$ of $CO_2$ released by a combustion module (Costech Analytical Technologies Inc., CA, USA) delivering the $CO_2$ resulting from combusting the sediment organic matter to the CRDS instrument.

**2.3 Assessment of carbon dioxide and methane air-seawater fluxes**

In February 2018, triplicate cores from vegetated and adjacent bare (about 5 m from the edge of the seagrass patch) sediments were collected from sites S1 and S2 and transferred to incubation chambers (Percival Scientific Inc., Perry, IA, USA) set at 25 ˚C and a 12 hours light (up to 70 µmol photons $m^{-2}$ $s^{-1}$) : 12 hours dark (12 h L : 12 h D) cycle to measure the greenhouse gas ($CO_2$ and $CH_4$) fluxes supported by these communities. Before measuring fluxes, the water overlying the

sediment inside the cores was carefully siphoned until only 5 mm of water remained over the sediment surface and fresh seawater was carefully siphoned in the core, to avoid disturbing the redoxcline, leaving a headspace of approx. 5 - 6 cm, and the cores were closed again with stoppers containing gas tight valves. The cores were left for one hour to allow for equilibration between the seawater and the headspace air. We then sampled 10 mL of air from each core using a syringe and injected the air sample in a cavity ring-down spectrometer (CRDS; Picarro Inc., Santa Clara, CA, USA) through a small

sample isotopic module extension (SSIM A0314, Picarro Inc., Santa Clara, CA, USA), which provided both the partial pressure and the isotopic carbon composition of the $CO_2$ and $CH_4$ in the air sample. One sample from each core was taken at the start (T0), after 12 hours of light (T1) and after 12 hours of dark (T2). The daily $CO_2$ and $CH_4$ fluxes were calculated from the difference between T0 and T2 taking into account the core surface area (µmol $m^{-2}$ $d^{-1}$). Before and after each sampling, two standards were measured (A: 750 ppm $CO_2$, 9.7 ppm $CH_4$, B: 250.5 ppm $CO_2$, 3.25 ppm $CH_4$).


**2.4 Effect of warming on carbon dioxide and methane air-seawater fluxes**

In March 2018, we collected eighteen vegetated and eighteen bare sediment cores from site S2 to evaluate the response of greenhouse gas fluxes to warming. The sampling was performed as described above. The cores were transferred to the Coastal and Marine Resources Core Lab (CMOR, KAUST, Saudi Arabia). Nine vegetated and nine bare sediment cores each

were placed in two aquaria with flow-through seawater set at *in situ* temperature (25 ˚C) and a 12 h L (up to 200 µmol photons $m^{-2}$ $s^{-1}$): 12 h D cycle. One aquarium was maintained at 25 ˚C over the entire duration of the experiment to serve as a control for temperature-independent variability in fluxes. The temperature in the second aquarium was increased at a rate of 1 ˚C $day^{-1}$ to allow for acclimatization of the vegetated and bare cores. $CO_2$ and $CH_4$ fluxes were measured at every 2 ˚C from 25 - 37 ˚C, with parallel measurements conducted on the cores maintained at 25 ˚C. After a one day acclimation period

at each new temperature, the cores were closed with the stoppers and transferred to incubation chambers (Percival Scientific

Inc., Perry, IA, USA) set at the target temperature for $CO_2$ and $CH_4$ flux measurements as described above. After the 24h measurements, the cores were returned to the aquaria. An additional core kept at each of the constant temperature and warming sets was sampled every four days (i.e. at 4 ˚C temperature intervals in the warming treatment) to analyze sediment composition. The cores used for fluxes estimates were opened after the final measurement (20 days since collection) to

estimate the plant biomass, analyze the sediment composition at the end of the experiment.

**2.5 Effect of darkness on carbon dioxide and methane air-seawater fluxes**

In May 2018, six vegetated and six bare sediment cores were collected from site S2 and kept at a constant 25 ˚C with a 24 hours dark cycle. Only during the measurements in the incubators, the cores were exposed to a 12 h L : 12 h D cycle,

allowing to compare fluxes with those measured in cores permanently maintained under the 12 h L : 12 h D photoperiod. $CO_2$ and $CH_4$ fluxes were measured after the first day of acclimation and then kept in the aquaria until signs of seagrass mortality started to become apparent, which occurred after one week in the dark. $CO_2$ and $CH_4$ fluxes were measured at alternate days as detailed above. At the end of the experiment (21 days since collection), the cores were opened and sampled to assess plant biomass, sediment composition.

**2.6 Measurements of carbon dioxide and methane air-seawater fluxes**

The concentration of $CO_2$ in the seawater after equilibrium was calculated based on the concentration of $CO_2$ in the headspace (ppm) measured by CRDS according to Sea et al. (2018) and Wilson et al. (2012):

$$[CO_2]_w = 10^{-6} \beta m_a p_{dry} \qquad (4)$$

where $\beta$ is the Bunsen solubility coefficient of $CO_2$ (mol $mL^{-1}$ $atm^{-1}$), $m_a$ is the $CO_2$ concentration measured in the headspace (ppm), and $p_{dry}$ is the atmospheric pressure of dry air (atm). The Bunsen solubility coefficient of $CO_2$ was calculated using Eq. 5:

$$\beta = H^{cp} \times (RT) \qquad (5)$$

where $H^{cp}$ is the Henry constant (mol $mL^{-1}$ $atm^{-1}$) calculated using the *marelac* R package (Soetaert et al., 2010). $R$ is the

ideal gas constant (0.082057459 atm L mol $mL^{-1}$ $K^{-1}$) and $T$ is the standard temperature (273.15 K).

The atmospheric pressure of dry air ($p_{dry}$) was calculated as follows:

$$p_{dry} = p_{wet}(1 - \%H_2O) \qquad (6)$$

where $p_{wet}$ is the atmospheric pressure of wet air. The Boyle's law was applied as gas was collected several times from the same core.

The concentration of dissolved $CO_2$ in seawater before equilibrium was then calculated using Eq. 7:

$$[CO_2]_{aq} = \frac{[CO_2]_w V_w + 10^{-6} m_a V_a}{V_w} \qquad (7)$$

where $V_w$ is the volume of seawater (mL) and $V_a$ is the volume of the headspace (mL). The units were then converted to nM:

$$[CO_2]_{aq} = \frac{10^9 \times p_{dry}[CO_2]_{aq}}{RT} \qquad (8)$$

The daily $CO_2$ fluxes were calculated from the difference between T0 and T2 taking into account the core surface area ($\mu$mol

$m^{-2}$ $d^{-1}$).

Daily $CH_4$ fluxes were estimated using the same calculations as for the $CO_2$ fluxes with the exception of the Bunsen solubility coefficient. The Bunsen solubility coefficient was calculated as a function of the seawater temperature and salinity following Wiesenburg and Guinasso (1979). The total $CO_2$ greenhouse-equivalent fluxes were calculated assuming $CH_4$ to have a greenhouse potential 28-fold greater than that of $CO_2$ per mol of C (Myhre et al., 2013).

### 2.7 Isotopic composition of carbon dioxide ($\delta^{13}C$-$CO_2$) and methane ($\delta^{13}C$-$CH_4$)

The isotopic signature of $CO_2$ and $CH_4$ produced during the incubations was estimated using Keeling plots following Garcias-Bonet and Duarte (2017). $\delta^{13}C$ of $CO_2$ and $CH_4$ produced in our incubations was extracted from the intercept of the linear regression between the inverse of the gas concentration ($ppm^{-1}$) and the isotopic signature measured from the discrete

samples in the CDRS instrument.

### 2.8 Data analysis

The data was analyzed for normality using the Shapiro-Wilk test. Mann-Whitney and t-test were used to test for differences in seagrass and sediment composition between sites and between vegetated and bare sediments, and ANOVA and Kruskal-

Wallis test were used to test for differences between vegetated and bare sediments and both sites. To assess differences in greenhouse gas fluxes between different *H. stipulacea* communities, differences in $CO_2$ and $CH_4$ fluxes were analyzed between sites and between vegetated and bare sediment by using Kruskal-Wallis test. Trends in the flux between the communities experiencing warming and the ones maintained at 25 ˚C, as well as in the isotopic signature of $\delta^{13}C$-$CO_2$ and $\delta^{13}C$-$CH_4$ over time were analyzed by linear regression. When assessing the effect of darkness on greenhouse gas fluxes, the

trend of $CO_2$ and $CH_4$ fluxes and their isotopic signatures were analyzed by linear regression. The statistical analyses were conducted in PRISM 5 (GraphPad Software, La Jolla, CA, USA) and JMP Pro 13.1.0 (SAS Institute Inc., Cary, NC, USA). The data is openly available from Burkholz et al. (2019b).

## 3 Results

### 3.1 Seagrass and sediment composition

Carbon, nitrogen (N), and phosphorus (P) concentrations in seagrass leaves were low, but they were 4- to 40-fold higher than vegetated and bare sediment concentrations (Table 1). Seagrass sampled in site S1 had the highest C, N and P concentrations in the leaves, while sediment C and P concentrations differed significantly between sites (ANOVA, $p < 0.05$ and Kruskal-Wallis, $p < 0.001$, respectively), with the highest C and the lowest P concentrations found in the sediment of S2 (Table 1).

There were no consistent differences in C, N and P concentration in bare and vegetated sediments (Table 1).

The sediments had high, but variable, carbonate concentrations, which differed between sites (Kruskal-Wallis, $p < 0.0001$; Table 1). The organic matter content was slightly higher in S2 than in S1, in both vegetated (t-test, $p < 0.0001$) and bare (t-test, $p < 0.001$) sediments (Table 1). Sediment bulk density was similar in both S1 and S2 sites, but vegetated sediments in S1 showed significantly lower bulk density compared to bare sediments (t-test, $p < 0.05$; Table 1). Seagrass biomass was

significantly higher in S2 compared to S1 (t-test, $p < 0.05$). The isotopic signature of sediment organic carbon ranged across sites from -15.77 ± 0.07 ‰, in vegetated sediments, to -16.36 ± 0.28 ‰, in bare sediments (Table 1). The carbon isotopic signature of seagrass leaves from the same location has been recently reported as -7.96 ± 0.27 ‰ by Duarte et al. (2018).

**3.2 Carbon dioxide and methane air-seawater fluxes**

The daily $CO_2$ flux was up to 6-fold higher in vegetated compared to bare sediments and tended to be generally higher in S2 compared to S1, where bare sediments showed net $CO_2$ uptake, although differences were not significant (Kruskal-Wallis, $p > 0.05$; Fig. 1A). At both sites, S1 and S2, the daily net $CH_4$ flux was 10- to 100-fold higher in vegetated compared to adjacent bare sediments with generally higher fluxes at S2 (Kruskal-Wallis, $p < 0.01$; Fig. 1B). The total $CO_2$ greenhouse-equivalent fluxes varied between sites and were higher in the vegetated compared to the bare sediments (Kruskal-Wallis, $p < 0.01$; Fig. 1C).

**3.3 Effect of warming on carbon dioxide and methane air-seawater fluxes**

The $CO_2$ fluxes in vegetated sediments increased greatly with warming ($R^2 = 0.38$, $p < 0.001$), but decreased over time when the community was maintained at 25 ˚C ($R^2 = 0.30$, $p < 0.01$; Fig. 2A, Table S1), shifting from sediments showing net $CO_2$ emission to net $CO_2$ uptake. Similar responses were observed in the bare sediments, where $CO_2$ fluxes increased with warming ($R^2 = 0.54$, $p < 0.0001$), while the community tended to shift over time from supporting net $CO_2$ emission to net $CO_2$ uptake when the maintained at 25 ˚C (Fig. 2B). The net $CO_2$ flux, i.e. the difference between the $CO_2$ fluxes in warming sediments and those at sediments maintained at 25 ˚C, increased significantly with warming in both the vegetated and the bare sediment ($R^2 = 0.74$, $p < 0.05$ and $R^2 = 0.91$, $p < 0.001$, respectively, Fig. 2C).

$CH_4$ fluxes declined over time when the sediments were maintained at 25 ˚C, both in vegetated ($R^2 = 0.43$, $p < 0.001$, Fig. 2D) and, less strongly, bare sediments ($R^2 = 0.24$, $p < 0.01$; Fig. 2E, Table S2). In contrast, $CH_4$ fluxes tended to increase with temperature in vegetated (Fig. 2D) and bare (Fig. 2E) sediments gradually warmed from 25 ˚C to 37 ˚C, although it was not significant ($p > 0.05$ and $p > 0.05$, respectively; Table S2). The net $CH_4$ fluxes, i.e. the difference between the $CH_4$ fluxes in sediments exposed to warming and those sediments at maintained at 25 ˚C, increased significantly over time (i.e. with warming) in vegetated ($R^2 = 0.69$, $p < 0.05$) but not in bare sediments ($p > 0.05$; Fig. 2F). An outlier in the vegetated sediment at 33 ˚C supporting extreme emissions ($CO_2$ flux of 55,170 µmol $CO_2$ m$^{-2}$ d$^{-1}$ and $CH_4$ flux of 699.8 $CH_4$ µmol m$^{-2}$ d$^{-1}$), was observed on day 14 in one of the replicates of the warming treatment where seagrass had died (Fig. 2A and D), and was excluded from the regression analyses reported above.

Despite $CO_2$ and $CH_4$ fluxes showing the same response to warming in both types of sediment, vegetated sediments held higher fluxes than bare sediments. The relationship between net $CO_2$ and $CH_4$ fluxes in bare vs. vegetated sediments showed that both bare and vegetated communities tended to act as net $CO_2$ sinks at 25 ˚C, but tended to act as $CO_2$ sources at warmer temperatures (Fig. 3A), whereas net $CH_4$ fluxes were 3- to 8-fold higher in vegetated compared to bare sediments. (Fig. 3B). The $CH_4/CO_2$ ratio declined in the vegetated sediments exposed to warming from 7 to 0.8 %. For $CO_2$ and $CH_4$ fluxes in vegetated sediments, the Q10 value for the temperature range 25-37 ˚C was 9 and 1.5, respectively, while the Q10 value for bare sediments was 13.8 and 4.2, respectively.

**3.4 Effect of darkness on carbon dioxide and methane air-seawater fluxes**

The vegetated sediment shifted over time from showing net $CO_2$ uptake to net $CO_2$ emission when maintained in the dark ($R^2 = 0.70$, $p < 0.05$), while the increase in the bare sediment was not significant ($p > 0.05$; Fig. 4A, Table S3). In contrast, when vegetated sediment was maintained at 25 ˚C at a 12 h L : 12 h D photoperiod, the community shifted from net $CO_2$ emission to net $CO_2$ uptake (Mann Whitney, $p < 0.05$; Fig. 5A). In bare sediments, $CO_2$ fluxes showed the same trend in cores maintained at 25 ˚C at a 12 h L : 12 h D photoperiod than under dark conditions (Fig. 5B).

When vegetated sediments were kept in the dark, net $CH_4$ fluxes decreased 5-fold over time ($R^2 = 0.99$, $p < 0.0001$; Fig. 4B, Table S4). However, the $CH_4$ fluxes did not differ significantly between vegetated cores maintained at 25 °C in the 12 h L : 12 h D photoperiod or in the dark (Mann Whitney, $p > 0.05$), showing the same trend of decreasing $CH_4$ fluxes (Fig. 5C). In the bare sediment, $CH_4$ fluxes in sediments kept in the dark were higher than those at 25 °C under a 12 h L : 12 h D photoperiod, with significant differences only observed on days 14 and day 20 (Mann Whitney, $p < 0.05$ and $p < 0.05$, respectively; Fig. 5D).

### 3.5 Isotopic composition of carbon dioxide ($\delta^{13}C\text{-}CO_2$) and methane ($\delta^{13}C\text{-}CH_4$)

The isotopic signature of the $\delta^{13}C\text{-}CO_2$ became heavier with warming in the bare sediment, increasing from $-22.36 \pm -4.97$ ‰ $\delta^{13}C$ at 25 °C to $-9.01 \pm 0.98$ ‰ $\delta^{13}C$ at 37 °C ($R^2 = 0.91$, $p < 0.001$), while the other treatments showed similar values over time, ranging from a minimum average of $-17.89 \pm 1.81$ ‰ to a maximum average of $-11.55 \pm 5.32$ ‰ $\delta^{13}C$ (Fig. 6A-D).

The isotopic signature of $\delta^{13}C\text{-}CH_4$ decreased over time in both vegetated and bare sediments, whether they were maintained at constant temperature or experienced warming (Fig. 6E-H). The isotopic signature in the vegetated sediment exposed to warming decreased significantly from $-50.8$ to $-54.06$ ‰ ($R^2 = 0.67$, $p < 0.001$).

The $\delta^{13}C$ isotopic composition of both $CO_2$ and $CH_4$ became heavier over time when the community was kept in the dark (Fig. 7), with a significant increase of $\delta^{13}C\text{-}CH_4$ in bare sediments ($R^2 = 0.94$, $p < 0.01$; Fig 7D).

## 4 Discussion

### 4.1 Carbon dioxide and methane air-seawater fluxes

The values reported for $CO_2$ and $CH_4$ fluxes varied greatly between the two sites studied here, with higher fluxes in the more organic sediments with higher biomass (S2). $CO_2$ and $CH_4$ fluxes were also highly variable over time in the studied site, as the first evaluation of fluxes in the same location delivered rates up to 100-fold above the rates of the second measurement one week later. Hence, organic matter availability along with temperature may account for the large variation in $CO_2$ and $CH_4$ fluxes. Additionally, the variability of $CO_2$ and $CH_4$ fluxes could also be supported by infaunal species present in the cores that were not recorded in this study. These trends were similar to results reported in previous studies, as a high variability between species and locations was found (cf. Table 1 in Garcias-Bonet and Duarte (2017)).

Even though there were some differences, carbon, nitrogen and phosphorus concentrations were generally similar, and they didn't seem to have an effect on $CO_2$ and $CH_4$ fluxes. Carbon, nitrogen and phosphorus concentrations were low compared to mean values (Carbon: $33.6 \pm 0.31$ % DW, nitrogen: $1.92 \pm 0.05$ % DW, phosphorus: $0.23 \pm 0.011$ % DW) reported for seagrass leaves by Duarte (1990). Serrano et al. (2018) explained the discrepancy between Red Sea data and global data with the extreme conditions in the Red Sea, such as low nutrient input and high temperatures, as well as a limited data set favoring high carbon stocks in the Mediterranean.

The results presented here add to those by Garcias-Bonet and Duarte (2017) to identify Red Sea seagrass communities as a significant source of $CH_4$. The presence of seagrass resulted in a higher organic matter supply to the sediments, favoring the presence of methanogens, which led to higher $CH_4$ fluxes compared to those fluxes supported in bare sediments (Barber and Carlson, 1993; Bahlmann et al., 2015), consistent with the up to 100-fold higher $CH_4$ fluxes supported by vegetated compared to bare sediments in this study. Additionally, higher fluxes in vegetated cores could be an indicator of direct effects resulting from the presence of seagrass, as vascular plants on land have shown to have varying effects on methane emissions caused by differences in biomass and gross photosynthesis (Öquist and Svensson, 2002).

Similar trends were also seen by Garcias-Bonet and Duarte (2017) who reported an increase in $CH_4$ fluxes with increasing organic matter content in Red Sea seagrass sediments. They reported organic matter contents in Red Sea seagrass sediments ranging from $2.33 \pm 0.07$ % (*Halodule uninervis*) to $12.42 \pm 0.23$ % (*Enhalus acoroides*), including a mixed meadow with *H. stipulacea* and *H. uninervis* showing a slightly higher organic matter content of $3.51 \pm 0.17$ % compared to vegetated
sediments at S2. Moreover, they found the highest $CH_4$ fluxes in meadows with the highest biomass, confirming our findings with higher fluxes in study site S2.

In terms of $CO_2$ equivalent greenhouse potential, only the bare sediment maintained at 25 ˚C seemed to act as a C sink over the experimental period, while the vegetated sediments, both maintained at 25 ˚C and exposed to warming, acted as sources of greenhouse gases. A sublethal disturbance, such as warming below the lethal threshold, can therefore lead to a shift of
seagrass ecosystems from acting as net sinks to net sources of greenhouse gases, as demonstrated experimentally here.

**4.2 Effect of warming**

Both $CO_2$ and $CH_4$ fluxes were higher in vegetated compared to adjacent bare sediments, indicating elevated remineralization rates in vegetated sediments as well as a higher susceptibility of seagrass sediment to increasing
temperatures. Vegetated sediments exposed to warming shifted from acting as a $CO_2$ sink to an increasingly intense source, while the $CO_2$ fluxes in vegetated sediments maintained at 25 ˚C decreased over time. Warming leads to an increase in both community photosynthesis and respiration, with respiration increasing at a faster rate (Harris et al., 2006) explaining the shift to a $CO_2$ source in sediments exposed to a thermal stressor. However, the fluxes maintained at 25 ˚C showed a net $CO_2$ uptake with a mean of $464.78 \pm 156.6$ $\mu$mol $CO_2$ $m^{-2}$ $d^{-1}$ (Table S1), while those reported in a mixed *Halodule* sp. and
*Halophila* sp. meadow in India showed a net $CO_2$ release (dry season: $1{,}190 \pm 1{,}600$ $\mu$mol $CO_2$ $m^{-2}$ $d^{-1}$, wet season: $18{,}400 \pm 8{,}800$ $\mu$mol $CO_2$ $m^{-2}$ $d^{-1}$; Banerjee et al., 2018). Both values reported were measured at higher temperatures (dry season: $30 \pm 0.68$ ˚C, wet season: $27.94 \pm 0.72$ ˚C, Banerjee et al., 2018) compared to our fluxes measured at 25 ˚C, also indicating that temperature might lead to higher fluxes. An initial high $CO_2$ flux measured on day 2 after sampling could be an indicator for the experienced disturbance due to sample collection and transportation even though we allowed the cores some time to
adapt.

Mean $CH_4$ fluxes at *in situ* temperature (25 ˚C) in vegetated sediments were lower than the mean value of $85.1 \pm 27.8$ $\mu$mol $CH_4$ $m^{-2}$ $d^{-1}$ reported for other seagrass meadows in the Red Sea (Garcias-Bonet and Duarte, 2017). In contrast, the community exposed to warming reached a maximum average $CH_4$ flux almost 4-fold higher than the community held at 25 ˚C, and showed a clear increase with warming, relative to sediments held at 25 ˚C. The increase in $CH_4$ fluxes with warming
was consistent with reports from Barber and Carlson (1993) for a *Thalassia testudinum* community in Florida Bay and Garcias-Bonet and Duarte (2017) for Red Sea seagrass communities, who reported higher $CH_4$ fluxes at higher temperatures. Additionally, previous research has shown that methanogenesis has a higher thermal dependence than respiration and photosynthesis (Yvon-Durocher et al., 2014) confirming the trends seen here with increasing fluxes at higher temperatures. We also reported a 10-fold decline in $CH_4$ fluxes over time for sediment communities maintained at 25 ˚C, which could be
attributable to increased sulfate reduction, reduced $CH_4$ production or a combination of both. Methane is produced under anoxic conditions in marine sediments, yet only a small portion is released, as $CH_4$ production by methanogens is compensated for by $CH_4$ oxidation by sulfate-reducing bacteria (Barnes and Goldberg, 1976). Similar to the trends seen in $CO_2$ fluxes, the decrease in $CH_4$ fluxes could be attributable to an initial stress response to the disturbance caused by sample collection and transportation. While reduced photosynthetic activity and a degradation in biomass could result in higher CH4
fluxes (Lyimo et al., 2018), the cores maintained at 25 ˚C might show the effect of healthy conditions.

Increasing water temperature led to a decrease in the $CH_4/CO_2$ ratio. While there was ~7 % of sequestered carbon released as $CH_4$ to the atmosphere in vegetated sediments at 25 ˚C (on day 2), it decreased to ~0.8 % in vegetated sediments at 37 ˚C. In contrast, Banerjee et al. (2018) reported ~1% of carbon being released as $CH_4$.

The isotopic composition of $CO_2$ in all treatments showed generally heavier isotopic signatures compared to previous reports of seagrass carbon (average $\delta^{13}C$ value of -7.73 ± 0.11 ‰ for Red Sea seagrass and -7.57 ± 0.15 ‰ for *H. stipulacea* in the Red Sea; Duarte et al., 2018), indicating various organic matter sources such as macroalgae blades (13.38 ± 0.3 ‰), mangrove leaves (26.58 ± 0.13 ‰) and seston (25.43 ± 0.42 ‰; Duarte et al., 2018). However, the mean $\delta^{13}C$ value of Red Sea seagrass sediments was reported to be $-13.36 ± 0.4$ ‰ (Garcias-Bonet et al., 2019a), similar to the results found in this study. Our chosen study sites were located in an enclosed lagoon with a high abundance of mangrove forests, leading to the conclusion that mangroves might be a major source of organic matter for our study sites. However, a recent study applying stable isotope mixing models found the major contributors to the organic matter in seagrass sediments in the Red Sea to be seagrass leaves and macroalgae blades, with contributions of 43 and 37 %, respectively (Garcias-Bonet et al., 2019a).

The isotopic signature of $CO_2$ released from bare sediments shifted with warming, suggesting a shift from seston, mangroves and macroalgae as the organic matter supporting respiration to seagrass carbon as the source of $CO_2$. In the vegetated cores, the isotopic composition of $CO_2$ stayed rather constant, indicating several sources of organic carbon with no clear shift, regardless of warming.

The isotopic signature of $CH_4$ in vegetated sediments confirmed its biogenic source as previous reports have shown that the isotopic signature of $CH_4$ from biogenic sources can range from -40 to -80 ‰, while the isotopic signature of $CH_4$ from geological and thermogenic sources ranges from -30 to -50 ‰ (Reeburgh, 2014), The isotopic composition of $CH_4$ in bare sediments was generally at the lower end of this range, with no clear shift with increasing temperature.

The isotopic composition of $CH_4$ can be determined by the production of $CH_4$ (methanogenesis) leading to lower $\delta^{13}C$ values and the oxidation of $CH_4$ (methanotrophy) leading to higher $\delta^{13}C$ values (Whiticar, 1990). Garcias-Bonet and Duarte (2017) reported fluctuations in the isotopic signature of $CH_4$ in Red Sea seagrass meadows, suggesting an indication for both processes. When exposed to increasing temperatures, we observed a shift to a lighter isotopic signature of $CH_4$ in vegetated sediments, thereby indicating an increasing $CH_4$ production by methanogens with warming.

**4.3 Effect of prolonged darkness**

Communities maintained at 25 ˚C and a 12 h L : 12 h D photoperiod showed continuous net $CO_2$ uptake, while the communities kept in the dark shifted, as expected, to a heterotrophic state, acting as a $CO_2$ source. The net $CO_2$ production corresponded to community respiration rates, while that at 12 h L : 12 h D photoperiod corresponded to the net community production.

We found, however, no effect of prolonged darkness on $CH_4$ fluxes, suggesting that the elevated $CH_4$ fluxes in vegetated sediments were not directly supported by fresh photosynthetic products, but rather by the elevated organic matter content in vegetated sediments compared to bare ones. These findings were in contrast to those reported by Lyimo et al. (2018) who found increased $CH_4$ fluxes under shading indicating that degradation of belowground biomass might have been the key factor related to increased $CH_4$ fluxes. However, they also reported varying results for different shading intensities, with low intensity having similar fluxes compared to their control group (Lyimo et al., 2018). In contrast, Öquist and Svensson (2002) found that photosynthesis might be regulating methane fluxes in a subarctic peatland ecosystem, with in lower photosynthesis resulting in lower methane fluxes.

### 4.4 Implications

Reports on greenhouse gas fluxes by seagrass ecosystems are limited (Oremland, 1975; Barber and Carlson, 1993; Alongi et al., 2008; Deborde et al., 2010; Bahlmann et al., 2015; Garcias-Bonet and Duarte, 2017; Banerjee et al., 2018; Lyimo et al., 2018), and no reports had been previously published on how increasing seawater temperatures might affect greenhouse gas fluxes by seagrass ecosystems. Blue carbon ecosystems have shown to turn into C sources when disturbances led to mortality (Macreadie et al., 2015; Lovelock et al., 2017; Arias-Ortiz et al., 2018), consistent with the very large $CO_2$ and $CH_4$ fluxes observed in one vegetated sediment where the seagrass died when warmed to 33 ˚C. However, even where seagrass remained alive, warming led to elevated greenhouse fluxes. Additionally, the elevated nutrient and high organic matter stock in seagrass meadows, which supports a 1.7-fold higher organic matter content than surrounding bare sediments, can promote an increase in $CO_2$ and $CH_4$ fluxes following disturbance (Gonsalves et al., 2011; Sotomayor et al., 1994). Our results suggest that this stock in seagrass sediments may be remineralized to support net greenhouse gas fluxes at the warmer temperatures reached and with further warming of the Red Sea. Hence, warming may, as other disturbances (Lovelock et al., 2017), shift seagrass ecosystems from net sinks to net sources of greenhouse gases, thereby providing a feedback mechanism that may contribute to further enhance global warming.

### 5 Conclusion

In summary, this study reports, for the first time, experimental evidence that warming leads to increased greenhouse gas ($CO_2$ and $CH_4$) fluxes in a *H. stipulacea* meadow in the Red Sea, and it may lead to seagrass meadows shifting from acting as sinks to sources of greenhouse gases. Increased fluxes at higher temperatures can be an indication of higher remineralization rates and a higher susceptibility of vegetated sediments to temperature. The elevated organic matter content, higher biomass and higher plant activity in vegetated sediments led to increased $CO_2$ and $CH_4$ fluxes in vegetated compared to bare sediments and a much steeper increase in $CO_2$ and $CH_4$ fluxes with warming. In addition, prolonged darkness led to an increase in $CO_2$ fluxes, while $CH_4$ fluxes decreased over time, also indicating organic matter to be the driver. However, we also found a high variability in fluxes over time indicating that other factors, such as infaunal species, could play a role as well. While current focus is on conserving blue carbon ecosystems from losses due to deteriorated water quality or mechanical damage, our results show that sublethal warming may also lead to emissions of greenhouse gases from seagrass meadows, contributing to a feedback between ocean warming and further climate change.

### Acknowledgements

This research was funded by King Abdullah University of Science and Technology through baseline and CARF funding to CMD. We thank Paloma Carrillo de Albornoz, Mongi Ennasri and Vijayalaxmi Dasari for their help with analyses. We also thank Katherine Rowe and the Coastal and Marine Resources Core Lab (CMOR) for their assistance during field work and CMOR for their support with the experimental set-up.

### Author contributions

NG-B, CMD and CB designed the project. CB collected the samples and conducted the experiments. NG-B, CMD and CB analyzed the results, CMD and CB wrote the first draft of the manuscript, and all authors contributed substantially to the final manuscript. All authors approved the final submission.

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

**TABLES**

**Table 1.** Summary of vegetated sediment, bare sediment and seagrass leaves characteristics in the study sites (S1 and S2). NA: Not available.

| | S1 | | | S2 | | |
|---|---|---|---|---|---|---|
| | Vegetated sediment | Bare sediment | Seagrass leaf | Vegetated sediment | Bare sediment | Seagrass leaf |
| C concentration (% DW) | 0.43 ± 0.05 | 0.41 ± 0.03 | 17.6 ± 2.72 | 0.55 ± 0.03 | 0.52 ± 0.02 | 15.32 ± 1.48 |
| N concentration (% DW) | 0.07 ± 0.01 | 0.12 ± 0.01 | 1.06 ± 0.17 | 0.08 ± 0.002 | 0.09 ± 0.01 | 0.94 ± 0.07 |
| P concentration (% DW) | 0.03 ± 0.001 | 0.03 ± 0 | 0.12 ± 0.01 | 0.02 ± 0 | 0.02 ± 0.001 | 0.11 ± 0.01 |
| Carbonate content (% DW) | 91.75 ± 0.56 | 91.65 ± 0 | NA | 82.61 ± 0.50 | 83.63 ± 0 | NA |
| Organic matter (% DW) | 2.45 ± 0.09 | 2.46 ± 0.16 | NA | 3.26 ± 0.03 | 2.95 ± 0.04 | NA |
| Bulk density (g cm$^{-3}$) | 1.15 ± 0.02 | 1.28 ± 0.03 | NA | 1.1 ± 0.07 | 1.2 ± 0.04 | NA |
| Seagrass biomass (g DW m$^{-2}$) | 60.87 ± 1.24 | NA | NA | 164.66 ± 20.54 | NA | NA |
| Sediment $\delta^{13}C-C_{org}$ (‰) | -15.77 ± 0.07 | -15.94 ± 0.1 | NA | -15.81 ± 0.13 | -16.36 ± 0.28 | NA |
| Seagrass leaf $\delta^{13}C-C$ (‰), extracted from Duarte et al. (2018) | | | | -7.96 ± 0.27 | | |

**FIGURES**

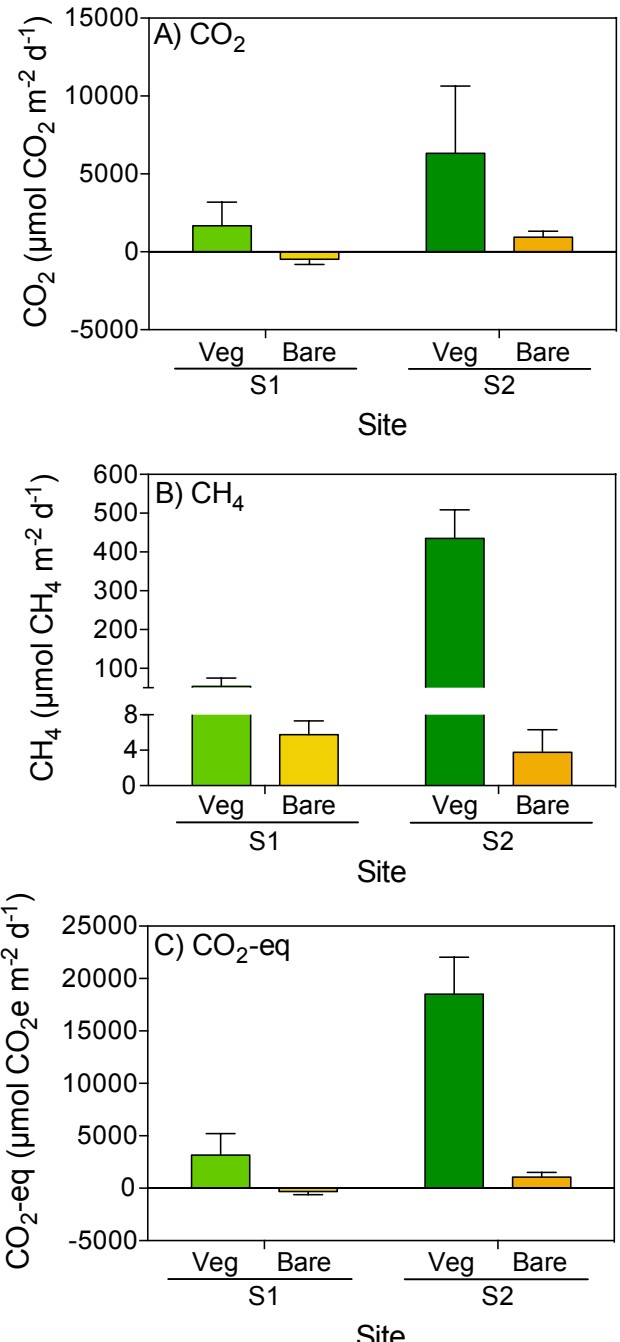

**Figure 1:** Mean + SE **(A)** $CO_2$, **(B)** $CH_4$, **(C)** $CO_{2\text{-eq}}$ fluxes in vegetated (green) and adjacent bare (yellow) sediments at two sites (S1 and S2) in the central Red Sea.

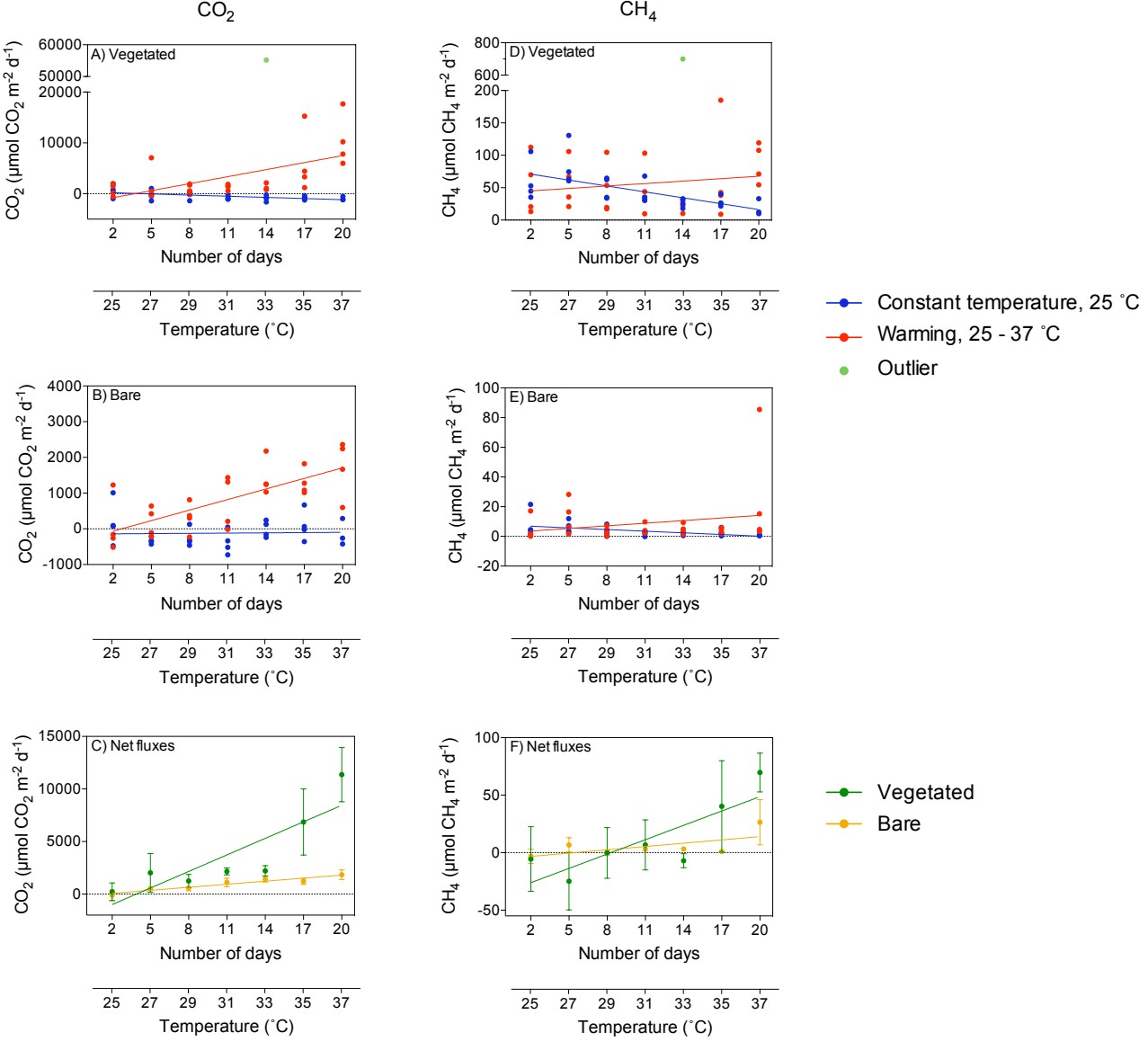

**Figure 2:** Mean ± SE $CO_2$ (left) and $CH_4$ (right) fluxes in **(A and D)** vegetated and **(B and E)** bare sediments. Symbols indicate each replicate of the community experiencing warming from 25 - 37 ˚C (red) and the community maintained at 25 ˚C (blue) over the experimental period (number of days since the onset of the experiment). An outlier at 33 ˚C in vegetated sediments in shown in green. **(C and F)** Mean ± SE $CO_2$ (C) and $CH_4$ (F) net fluxes in vegetated (green) and bare (yellow) sediments over the experimental period (number of days since the onset of the experiment). The second x-axis indicates the experimental temperature for the community exposed to warming from 25 - 37 ˚C. Lines represent a fitted linear equation.

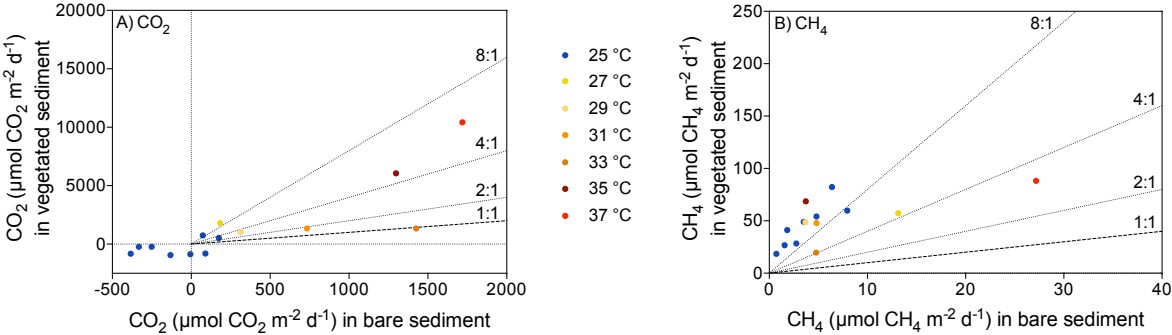

**Figure 3:** Relationship between vegetated and bare sediments for **(A)** $CO_2$ and **(B)** $CH_4$ fluxes. Symbols indicate different temperatures ranging from 25 - 37 °C, the dashed line indicates line 1:1, and dotted lines show lines 2:1, 4:1 and 8:1.

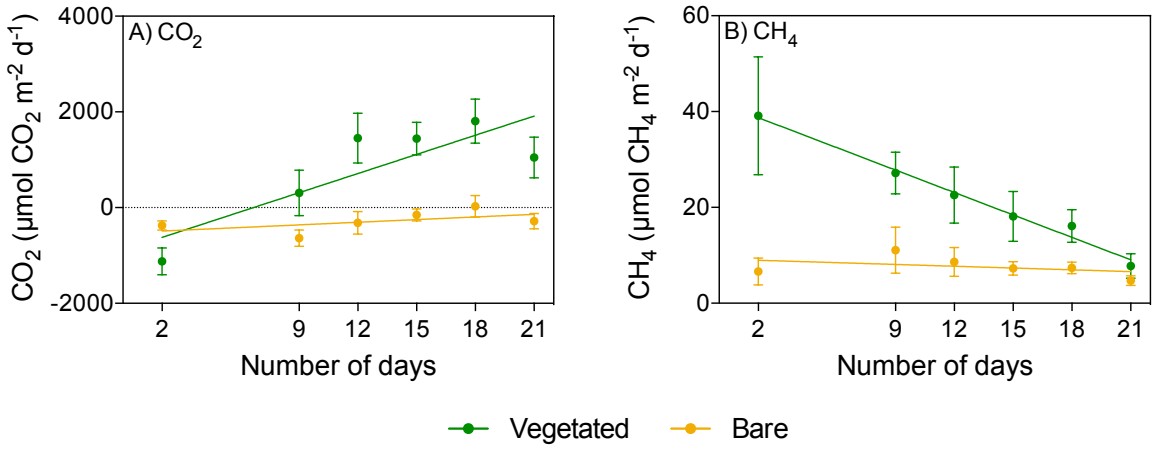

**Figure 4:** Mean ± SE **(A)** $CO_2$ and **(B)** $CH_4$ fluxes in vegetated (green) and bare (yellow) sediments of communities exposed to prolonged darkness over the experimental period (number of days since the onset of the experiment).

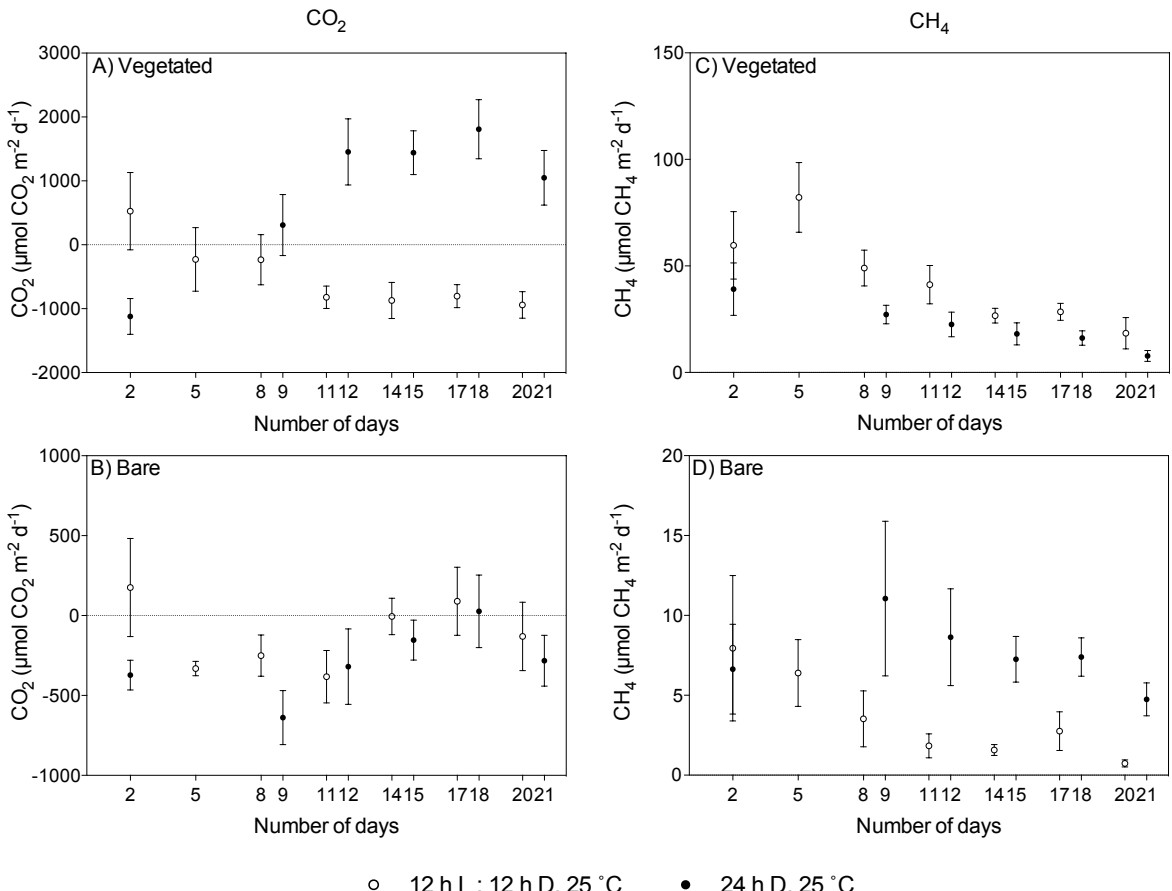

**Figure 5:** Comparison of mean ± SE $CO_2$ (left) and $CH_4$ (right) fluxes in **(A and C)** vegetated and **(B and D)** bare sediments maintained at 25 ˚C and a 12 h L:12 h D photoperiod (white) and communities kept at 25 ˚C and a 24 h D period (black) over the experimental period (number of days since the onset of the experiment). Dots indicate mean values and error bars indicate standard error of the mean.

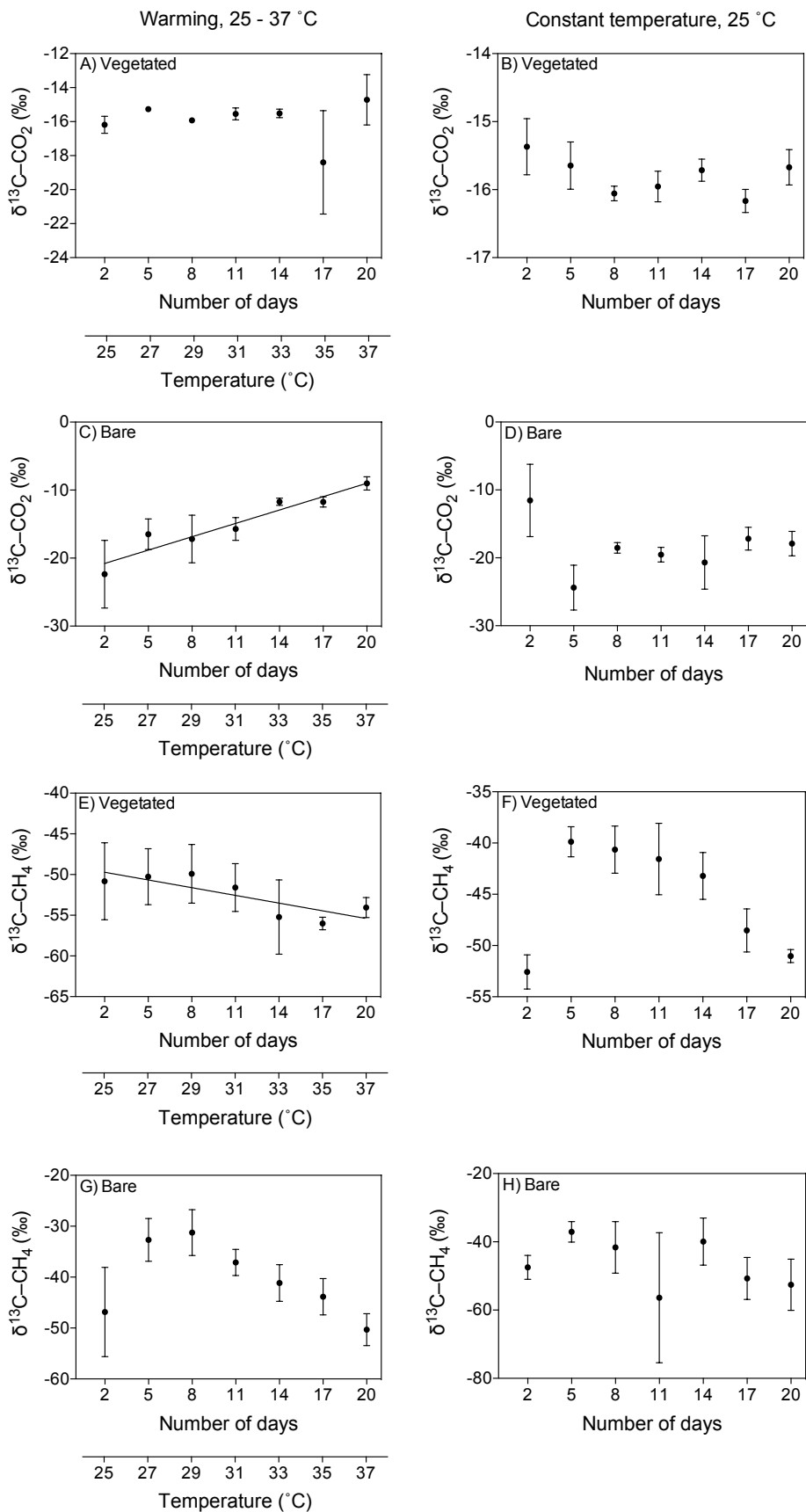

**Figure 6:** Mean ± SE isotopic signature of $CO_2$ ($\delta^{13}C\text{-}CO_2$) and $CH_4$ ($\delta^{13}C\text{-}CH_4$) in the communities experiencing warming from 25 - 37 ˚C (left) and the communities maintained at 25 ˚C (right). **(A-D)** $\delta^{13}C\text{-}CO_2$ is shown for the vegetated (A and B) and bare (C and D) sediments over the experimental period (number of days since the onset of the experiment). **(E-H)**

$\delta^{13}$C-CH$_4$ is shown for the vegetated (E and F) and bare (G and H) sediment over the experimental period. The second x-axis indicates the temperature increase for the community experiencing warming.

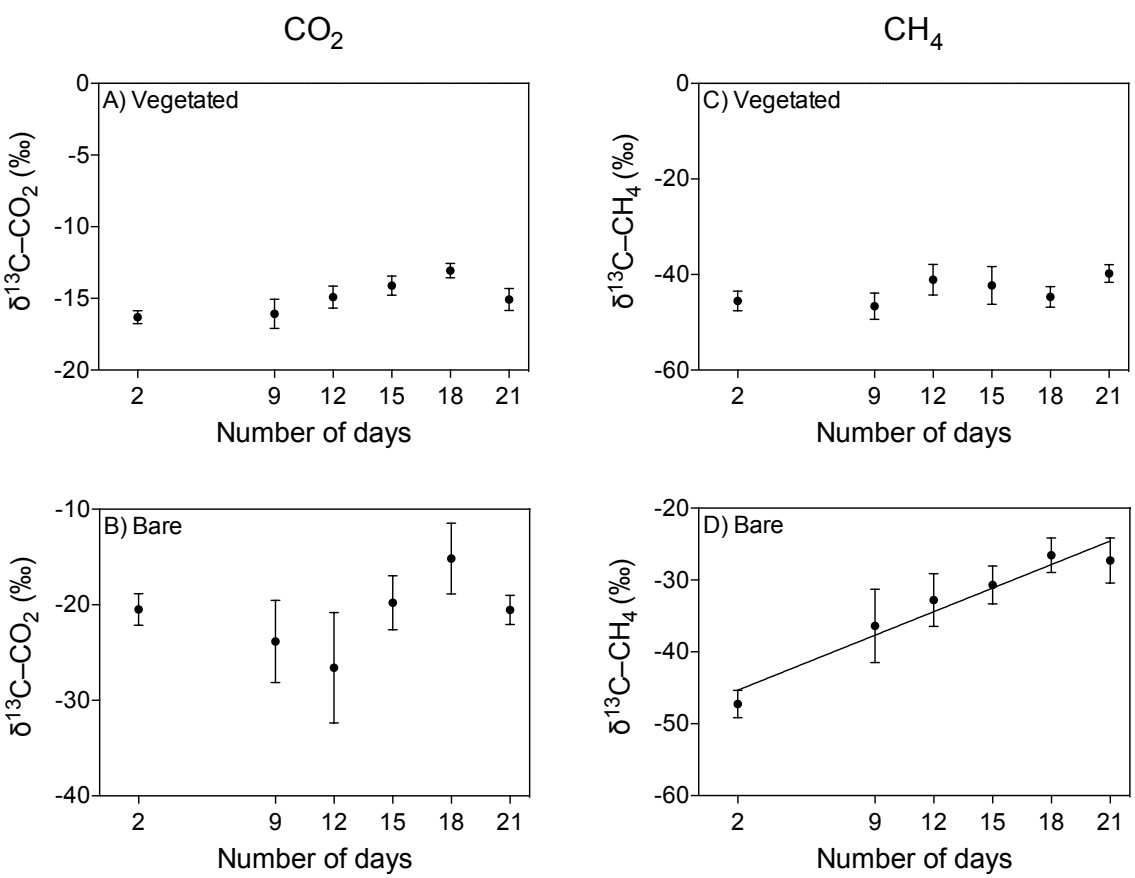

5     **Figure 7:** Mean ± SE isotopic signature of CO$_2$ ($\delta^{13}$C-CO$_2$, left) and CH$_4$ ($\delta^{13}$C-CH$_4$, right) in **(A and C)** vegetated and **(B and D)** bare sediments exposed to prolonged darkness over the experimental period (number of days since the onset of the experiment).

