# Peer review of "Warming enhances carbon dioxide and methane fluxes from Red Sea seagrass (*Halophila stipulacea*) sediments"

_Biogeosciences, 2019_

## Referee Comment (RC1) · Anonymous Referee #1 · 21 Aug 2019

This manuscript presents the results of a study in which experiments on impacts of warming and prolonged darkness on CO2 and CH4 fluxes are conducted in seagrass ecosystems of the Red Sea. Results show upward shifts in carbon dioxide and methane fluxes with warming and in the dark with a few exceptions under varied experimental conditions. Though it is known that a rise in temperature would increase metabolic rates the present set of results confirm thus driven elevated CO2 and CH4 fluxes for seagrass meadows in the Red sea. These results are of significance to understanding and quantifying the forcings and feedbacks of climate system. The Results and Discussion Sections were presented well but I found it difficult to follow some statements in Introduction section. Besides there is need to improve clarity to Material and

[Figure]

Methods Section by furnishing more details. Specific comments are given below:

Page 2 Lines 8-10: "where autotrophic communities [net community production (NCP) > respiration (R)] act as a sink for carbon dioxide ($CO_2$), while heterotrophic communities [net community production (NCP) < respiration (R)] act as a source of $CO_2$ (Duarte et al., 10 2011)." - Why not make it simple? Say 'where net community production (NCP) > respiration (R)] the system becomes a sink for carbon dioxide ($CO_2$).'?

Line 38-39: "warming at higher rates than those of the global ocean" - at what rates? Specific information will be helpful.

Page 3 Line 30: "Once the cores were opened, the first 10 cm of the sediment and the plant biomass were collected and dried" - Is this biomass picked from the same sediment core or was it collected separately? In fact Line 24 says that sediments were collected to a depth of 10 cm. If yes, then what is 'the first 10 cm' in Line 30? This is confusing.

Page 4 Line 16 "triplicate cores from vegetated and adjacent bare (about 5 m from the edge of the seagrass patch)" - Can a sample just 5 m away from the edge of the sea-grass patch be true representative of 'bare' sediment? Table 1 shows that sediment characteristics between vegetated and bare sediments of S2 are nearly the same but for marginal high organic matter content in the former. Only the other differences expected under these circumstances could be nature and density of microbes on which 'respiration rate' essentially depends on!

Line 22: "We then sampled 10 mL of air from each core using a syringe". Which replacement air was used to put into headspace each time 10 ml of air sample was drawn and how?

Line 30-31: "In March 2018, we collected eight vegetated and eight bare sediment cores from site S2 to evaluate the response of greenhouse gas fluxes to warming." - This sentence says eight cores each from vegetated and bare sediments. But how

the number became NINE each in Lines 32-33 ("Nine vegetated and nine bare sediment cores were placed in each two aquaria")? Also what is 'were placed in each two aquaria'? Did they mean 'were placed separately in two aquaria'? Since they collected 8 cores each from vegetative and bare sediment zones I would expect them to place 4 cores from each zone (total 8 cores) in each aquarium! Their write-up is confusing!!! Or more clarity is needed in presentation.

Page 6: Lines 24-25: "Carbon, nitrogen, and phosphorus concentrations in seagrass leaves were low, but C, N and P leaf concentrations were 4- to 25 40-fold" - Did the authors mean 'vegetative sediments or sediments for seagrass leaves'?

Page 8: Line 8: "ranging from a minimum average of -11.55 ± 5.32 ‰ to a maximum average of -17.89 ± 1.81 ‰ $\delta$13C" – are minimum and maximum interchanged? Please note that these values are bear negative sign.

Lines 20-21: "$CO_2$ fluxes were also 10-fold higher in vegetated compared to adjacent, but bare sediments, indicating elevated microbial remineralization rates in vegetated sediments." Rewrite as words are repetitive and a bit confusing too. Given this statement minimal microbial description of these sediments will be very helpful.

Lines 34-35: "Mean $CH_4$ fluxes at in situ temperature (25 ËŽC) in vegetated sediments were lower than the mean value of 85.09 ± 27.80 35 $\mu$mol CH4 m-2 d-1" - Caution needs to be exercised when expressing flux values to the second decimal. This is unnecessary given the uncertainties associated with flux estimates in general and large mean deviation in this particular case. ***

---

## Referee Comment (RC2) · Anonymous Referee #2 · 17 Sep 2019

Review report of the paper entitled "Warming enhances carbon dioxide and methane fluxes from Red Sea seagrass (Halophila stipulacea) sediments" submitted by Celina Burkholz et al to Biogeosciences

Seagrass meadows, saltmarshes and mangroves are the "hot spot" blue carbon sinks and highly efficient in long-term carbon storage in the coastal marine ecosystems. Deterioration of these natural marine ecosystems through anthropogenic perturbation could change their carbon sinks efficiency and may contribute to climate change through re-emissions of locked carbon dioxide and other greenhouse gases. The warming-dependent emissions of metabolic GHG in the coastal ecosystems are likely

to vary for coastal sediment of different geological origin. It is expected that in situ temperature increase is particularly important in seagrass meadows which have high carbon sequestration with long-term storage capacity and trap organic matter from external sources. The extreme conditions in the Red Sea i.e. slower seagrass growth due to nutrient limitation and greater microbial degradation of soil organic carbon because of high temperature could be related to low Corg storage in the sediment compared to temperate meadows. This paper reports the results of the study on the response of sediment collected from two H.stipulacea meadows, S1 (22ËŽ56.775'N, 38ËŽ52.677'E) and S2 (22ËŽ54.742'N, 38ËŽ53.848'E) at Al Kharar, a lagoon on the Saudi coast of the central Red Sea in February-March 2018, in terms of air-seawater fluxes of $CO_2$ and $CH_4$ along with their isotopic signature from Red Sea meadows compared to for gradual increase of temperature from 25 ËŽC to 37 ËŽC and prolonged darkness. However, I had difficulties to understand certain sections of the manuscript including the methodology and discussion. Furthermore, the author did not address appropriately other important elements particularly the redox conditions, anthropogenic pressure in and around the the ecosystems (Seagrass and bare soil). The sediment in the present study is of carbonate (82.61-91.75% ) dominated. What constitutes the rest fraction (9-18%) of the sediment? It does not represent all types of sediment of different geological origin in Al Kharar lagoon. Previous reports (Serrano et al. 2018 Scientific Reports, 8:15037) indicate that soils in seagrass meadows in Saudi Arabia, Central Red Sea. are mainly constituted of clay and silt particles (37 ± 0.7% on average), with a relatively high abundance of very fine sands (21 ± 0.4%) compared to fine sands (16 ± 0.4%), medium sands (12 ± 0.3%) and coarse sands (14 ± 0.7%). Youssef & El-Sorogy, 2016 ( Arab J Geosci 9:474) showed that sediment textures consist of mud, gravelly sand, and sandy mud and organic matter in the sediment may also be derived from Mangroves which are common in Al-Kharrar lagoon between latitude 22° 45 and 23° 00 N and longitude 39° 00 and 38° 45 E. Why you have collected sediment from two H.stipulacea meadows only? Why the sediment of two H.stipulacea meadows showed high bulk density (1.1-1.28 g cm-3 ) relative to silt-clay sediment (

**BGD**

1.05 g cm-3 ) in seagrass meadows in Saudi Arabia, Central Red Sea Serrano et al. 2018 observed. The bulk density of carbonate sediment (carbonate 0.7 g/cm3) should be even lower than that of sediment containing mainly sand (1.586 g cm-3) and clay-silt ( 1.0 to 1.6 g/cm3). Organic carbon content of both vegetated (0.43 -0.55%) and bare (0.41 - 0.52%) carbonate sediment of H. stipulacea meadows is greater than the value reported for clay-silty sediment of other seagrass meadows (0.33%, Serrano et al. 2018 ). Why the production of H.stipulacea is high at the study site S2 compared to S1 and other seagrass meadows along the Saudi coast even though there was almost no available nutrients (N & P), and no difference in nutrients between S1 and S2. If seagrass detritus is the significant source of soil organic carbon, what is the source of organic carbon for bare sediment? Why vegetated and bare sediment of S1 showed no difference in soil organic carbon content? The $\delta$ 13 C-Corg showed depletion of 13C in comparison to leaf and both vegetated and bare sediment showed no significant difference of $\delta$ 13 C-Corg. Moreover, given $\delta$ 13C value of -7.96 $\pm$ 0.27 0/00 is lower than that of literature value for C4 plants -14 0/00. This needs clarifications. Use stable isotope mixing model to determine the actual contribution of seagrass on organic carbon content of the meadow sediment. $CO_2$ fluxes were also 10-fold higher in vegetated compared to adjacent bare sediments, indicating elevated microbial remineralization rates in vegetated sediments. P 4: Before measuring fluxes, why the water inside the cores was replaced by fresh seawater ? Is it accumulated pore water ? Replacement of pore water saturated with $CO_2$ and $CH_4$ by fresh sea water may result serious error! Sediment core was incubated at one particular temperature and $CO_2$ and $CH_4$ concentrations were measured in 10 ml of head space air samples from each core at 0, 12hrs (light) and 12 hrs (dark). Considering, efluxes of 10,422 $\mu$mol $CO_2$ m-2 d-1 and 88.11 $\mu$mol $CH_4$ m-2 d-1, simple calculation shows after 12 hrs increased amount of $CO_2$ and $CH_4$ should be 37 $\mu$mol and 0.31 $\mu$mol, respectively. But standards (A: 750 ppm $CO_2$, 9.7 ppm $CH_4$, B: 250.5 ppm $CO_2$, 3.25 ppm $CH_4$) used before each (?) measurement of samples are all above those values. A plot of observed mixing ratio of $CO_2$ and $CH_4$ versus temperature should be given for better understanding.

[Figure]
The mean ratio of CH4/CO2 was found to be ∼0.008 which seems to be higher than the previous value reported for seagrass meadows (Halodule uninervis, Halodule pinifolia, Halophila ovalis, Halophila ovata, and Halophila beccarii ), Chilika Lagoon ( PLoS ONE 13(10): e0203922. https://doi.org/10.1371/journal.pone.0203922). The organic matter content was higher in S2 (vegetated 0.55%) than in S2 (bare 0.52%) by 5% . Is it below the error limit of its determination? How do you explain 6-fold CO2 and 100-fold CH4 greater emission in S2 (vegetated) than in S2 (bare) ? Provide a plot of observed CH4 concentration versus temperature . The Fig 2 D and E shows large scatter of data and drawing those straight lines have no use. The CH4 fluxes declined over time when the sediments were maintained at 25 ËŽC, both in vegetated ( Fig. 2D) and, less strongly, bare sediments. In contrast, CH4 fluxes tended to increase with temperature in vegetated (Fig. 2D) and bare (Fig. 2E) sediments gradually warmed from 25 ËŽC to 37 ËŽC, although it was not significant. Since the in situ redox condition of both water and sediment was not maintained during experiment that may affect the equilibrium between counteracting microbial processes of production and oxidation in the sediment. The study on the response of benthic net methane concentrations to higher temperatures needs also to take into account methane production rates, Q10 values, and community sizes of methanogens and methanotrophs in seagrass sediments . I believe that the manuscript needs significant revision before being considered for resubmission. Specific comments Abstract: In the first sentence please mention if Seagrasses are net source of CO2 and CH4. Line 17: "We detected distinct differences between vegetated and unvegetated sediments, with the vegetated sediments supporting 6-fold higher CO2 fluxes, and 10- to 100-fold higher CH4 fluxes" This is a confusing statement. What are the conditions for the high flux was not mentioned? Methodology "Assessment of carbon dioxide and methane air-seawater fluxes" This section is not clear. Please clarify the total number of samples collected from each core. Was there a periodic collection? Did the cores have seagrass in the top surface? What was the depth of the core sample? What were the criteria for fixing the light condition at 70 $\mu$mol photons m-2 s-1 and 200 $\mu$mol photons m-2 s-1 at different incubation conditions? "The temperature in the second aquarium was increased at a rate of 1 ËŽC day-1." Why this was done? Is this a natural increase (with 1day time) for the physiological adaptation by of seagrass? I wander how they have measured water-air flux from a system enclosed with sediment- water-air. Lots of Q10 values are available in the literature. Do those values agree with the present report? Results: The first sentence needs to be revised. Page 6 line 28: "There were no consistent differences in C, N and concentration in bare and vegetated sediments (Table 1)." Does it indicate limited influence of additional carbon storage in the seagrass sediment? "The mean C, N content is significantly lower in the seagrass leaf (Table 1) compared to global average carbon content of 35% (Duarte 1990; Fourqurean et al. 2012). There are several other seagrass sps apart from Halophila sp. in Kharar lagoon with considerable spatial variation sediment composition. Those Sps should also directly or indirectly affect the sediment composition. The sediment in the present study is manly composed of coral sand. Is it possible for the occurrence of such high concentration of OM in the sediment with no clay and silt? Page 6 Line 38: "The daily $CO_2$ flux was up to 6-fold higher in vegetated compared to bare sediments, and tended to be generally higher in S2 compared to S1, where bare sediments showed net $CO_2$ uptake, although differences were not significant" This result need to be discussed in details in the discussion section. Page 7 Line 21: "Despite $CO_2$ and $CH_4$ fluxes showing the same response to warming in both types of sediment, vegetated sediments held higher fluxes than bare sediments. The relationship between net $CO_2$ and $CH_4$ fluxes in bare vs. vegetated sediments showed that both bare and vegetated communities tended to act as net $CO_2$ sinks at 25 ËŽC, but tended to act as $CO_2$ sources at warmer temperatures (Fig. 3A), whereas net $CH_4$ fluxes were 3- to 8-fold higher in vegetated compared to bare sediments. (Fig. 3B)." The organic carbon contents are comparable for seagrass and bare soil. Do the result indicate higher susceptibility of the seagrass soil carbon at a higher temperature? Apart from regulation by Seagrass, $CH_4$ and $CO_2$ efflux depend on the redoxcline which has not been highlighted in relation to the abundance of methanogens and methanotrops. In methodology of the incubation experiment it is

mentioned that "….the water inside the cores was replaced by fresh seawater from the aquaria leaving a headspace of approx. 5 - 6 cm….". Did this replacement change the redox condition of surface water? Discussion This section is inadequate and the observed results need more detailed discussions about the variations CO2 and CH4 fluxes at natural, high temperature and low light conditions. "Similar trends were seen by Garcias-Bonet and Duarte (2017) who reported an increase in CH4 fluxes with increasing organic matter content in Red Sea seagrass sediments." In the present study lower CO2 and CH4 fluxes were recorded from adjacent bare sediments with comparable Organic C content. Page 8 Line 24: "….while the CO2 fluxes in vegetated sediments maintained at 25 ËŽC decreased over time." Why it was high initially and decreased gradually? explain. Line 30: "The presence of seagrass resulted in a higher organic matter supply to the sediments, favoring the presence of methanogens, which led to higher CH4 fluxes compared to those fluxes supported in bare sediments (Barber and Carlson, 1993; Bahlmann et al., 2015), consistent with the up to 100-fold higher CH4 fluxes supported by vegetated compared to bare sediments in this study." In the present study, soil organic C content in vegetated and adjacent bare sediment are comparable but CH4 fluxes are significantly different. Does the results (high emissions at S1 and S2) indicated direct CH4 emission from Seagrass? (pls see:Quist and Svensson, 2002, Vascular plants as regulators of methane emissions from a subarctic mire ecosystem, jgr, 107, NO. D21, 4580 and others) Effect of warming Page 8, Line 26 : "However, the fluxes maintained at 25 ËŽC were about 3-fold below those reported in a mixed Halodule sp. and Halophila sp. meadow in India (980 $\mu$mol CO2 m-2 d-1; Banerjee et al., 2018)." Is this the mean flux (the present value) or the least value recorded or the values represent during the end of the incubation at 25 ËŽC? In the previous sentence it was mentioned as "CO2 fluxes in vegetated sediments maintained at 25 ËŽC decreased over time"

Page 8, Line 40: "We also reported a 10-fold decline in CH4 fluxes over time for sediment communities maintained at 25 ËŽC, which could be attributable to increased sulfate reduction, reduced CH4 production or a combination of both. Methane is produced

under anoxic conditions in marine sediments, yet only a small portion is released, as CH4 production by methanogens is compensated for by CH4 consumption by sulfate-reducing bacteria (Barnes and Goldberg, 1976)." The soil redox conditions were not measured in this study. Please show some evidence of enhanced soil anoxicity which may have increased SO4 reduction. What could be the other reason/s of reduced CH4 production.

Conclusion

This section may be rewritten after revising the discussion section with few more synthesizing statements. Figure 2: what does the blue dots represent? Figure 3: ….. the dashed line indicates line 1:1, and dotted lines show lines 2:1, 4:1 and 8:1. Need to ve detailed

.

---

## Author Comment (AC1) · 28 Oct 2019

Authors' response

RC: Comments from referees/public, AR: Authors' response, AC: Authors' changes in manuscript

ANONYMOUS REFEREE #1
1) RC: This manuscript presents the results of a study in which experiments on impacts of warming and prolonged darkness on CO2 and CH4 fluxes are conducted in seagrass ecosystems of the Red Sea. Results show upward shifts in carbon dioxide and methane fluxes with warming and in the dark with a few exceptions under varied experimental conditions. Though it is known that a rise in temperature would increase metabolic rates the present set of results confirm thus driven elevated CO2 and CH4 fluxes for seagrass meadows in the Red sea. These results are of significance to understanding and quantifying the forcings and feedbacks of climate system. The Results and Discussion Sections were presented well but I found it difficult to follow some statements in Introduction section. Besides there is need to improve clarity to Material Methods Section by furnishing more details.

AR: We thank the reviewer for the constructive comments. We have made some changes to the manuscript to improve clarity.

Specific comments are given below.

2) RC: Page 2 Lines 8-10: "where autotrophic communities [net community production (NCP) > respiration (R)] act as a sink for carbon dioxide (CO2), while heterotrophic communities [net community production (NCP) < respiration (R)] act as a source of CO2 (Duarte et al., 10 2011)." - Why not make it simple? Say 'where net community production (NCP) > respiration (R)] the system becomes a sink for carbon dioxide (CO2).'?

AR: In fact, the statement was in error, besides complex. The sentence now reads as follows, which is a simpler, and most importantly, correct statement.

AC: Page 2, line 7-9: Ecosystem metabolism can also be a source of greenhouse gases, depending on the metabolic balance of the community, where autotrophic communities [net community production (NCP) > 0] act as a sink for carbon dioxide (CO2), while heterotrophic communities [net community production (NCP) < 0] act as a source of CO2 (Duarte et al., 2011).

3) RC: Line 38-39: "warming at higher rates than those of the global ocean" - at what rates? Specific information will be helpful.

AR: We thank the reviewer for pointing out the missing information. We have added the rates as requested.

AC: Page 2, line 38-40: The Red Sea ranks as the warmest sea in the world, with summer seawater temperatures reaching 35 ËŽC, and is warming at higher rates (0.17 $\pm$ 0.07 °C decade$-1$, Chaidez et al., 2017) than those of the global ocean (0.11 °C decade$-1$, Rhein et al., 2013).

4) RC: Page 3 Line 30: "Once the cores were opened, the first 10 cm of the sediment and the plant biomass were collected and dried" - Is this biomass picked from the same sediment core or was it collected separately? In fact Line 24 says that sediments were collected to a depth of 10 cm. If yes, then what is 'the first 10 cm' in Line 30? This is confusing.

AR: We agree with the reviewer's concern that this part can be confusing. We have edited the section accordingly. Regarding line 24, the cores were not taken at a depth of 10cm, the cores were pushed 10 cm into the sediment. We have added additional information for clarification. AC: Page 3, line 18-20: Two H.stipulacea meadows at a depth of 2-3 m, S1 (22 ÌŁ56.775'N, 38 ÌŁ52.677'E) and S2 (22 ÌŁ54.742'N, 38 ÌŁ53.848'E), were chosen to represent a range of organic matter content in the sediment, selected to evaluate greenhouse gas fluxes. Page 3, line 30-31: Once the cores were opened, the first 10 cm of the sediment and the plant biomass from the same cores were collected and dried.

5) RC: Page 4 Line 16 "triplicate cores from vegetated and adjacent bare (about 5 m from the edge of the seagrass patch)" - Can a sample just 5 m away from the edge of the seagrass patch be true representative of 'bare' sediment? Table 1 shows that sediment characteristics between vegetated and bare sediments of S2 are nearly the same but for marginal high organic matter content in the former. Only the other differences expected under these circumstances could be nature and density of microbes on which 'respiration rate' essentially depends on!

AR: We have chosen a distance of 5 m to show the difference between the absence and presence of seagrass. A further location would have implied a variation of many other factors (depth, sediment type, etc) that could have affected the results. By having similar sediment conditions, we can imply that differences can be caused by the presence/absence of seagrass biomass.

6) RC: Line 22: "We then sampled 10 mL of air from each core using a syringe". Which replacement air was used to put into headspace each time 10 ml of air sample was drawn and how?

AR: There was no replacement air used to add to the headspace. We followed the same methodology described in Garcias-Bonet et al. (2017) and Sea et al. (2018): First, the water inside the cores was replaced by fresh seawater leaving a headpsace, and the cores were closed again with stoppers containing gas tight valves. The cores were left for one hour to allow for equilibration between the seawater and the headspace air. We then sampled 10 mL of air from each core using a syringe and injected the air sample in a cavity ring-down spectrometer through a small sample isotopic module extension (SSIM A0314, Picarro). One sample from each core was taken at the start (T0), after 12 hours of light (T1) and after 12 hours of dark (T2).

7) RC: Line 30-31: "In March 2018, we collected eight vegetated and eight bare sediment cores from site S2 to evaluate the response of greenhouse gas fluxes to warming." - This sentence says eight cores each from vegetated and bare sediments. But how the number became NINE each in

AR: We thank the reviewer for pointing out the error. We have edited the sentence as follows:

AC: Page 4, line 32-33: In March 2018, we collected eighteen vegetated and eighteen

bare sediment cores from site S2 to evaluate the response of greenhouse gas fluxes to warming.

8) RC: Lines 32-33 ("Nine vegetated and nine bare sediment cores were placed in each two aquaria")? Also what is 'were placed in each two aquaria'? Did they mean 'were placed separately in two aquaria'? Since they collected 8 cores each from vegetative and bare sediment zones I would expect them to place 4 cores from each zone (total 8 cores) in each aquarium! Their write-up is confusing!!! Or more clarity is needed in presentation.

AR: We share the reviewer's concern that this phrasing might have been confusing. We have changed the sentences as follows:

AC: Page 4, line 34-36: Nine vegetated and nine bare sediment cores each were placed in two aquaria with flow-through seawater set at in situ temperature (25 ËŽC) and a 12 h L (up to 200 $\mu$mol photons m-2 s-1): 12 h D cycle.

9) RC: Page 6: Lines 24-25: "Carbon, nitrogen, and phosphorus concentrations in seagrass leaves were low, but C, N and P leaf concentrations were 4- to 25 40-fold" - Did the authors mean 'vegetative sediments or sediments for seagrass leaves'?

AR: We agree with the reviewer that this is not clear, we have added the missing information that we were referring to both sediments, vegetated and bare. AC: Page 6, line 26-27: Carbon, nitrogen (N), and phosphorus (P) concentrations in seagrass leaves were low, but they were 4- to 40-fold higher than vegetated and bare sediment concentrations (Table 1). 10) RC: Page 8: Line 8: "ranging from a minimum average of -11.55 $\pm$ 5.32 ‰ to a maximum average of -17.89 $\pm$ 1.81 ‰ $\delta$13C" – are minimum and maximum interchanged? Please note that these values are bear negative sign.

AR: We thank the reviewer for pointing out this mistake. The sentence was corrected accordingly.

AC: Page 8, line 9-12: The isotopic signature of the $\delta$13C-CO2 became heavier with

warming in the bare sediment, increasing from -22.36 ± -4.97 ‰ $\delta$13C at 25 ËŽC to -9.01 ± 0.98 ‰ $\delta$13C at 37 ËŽC (R2 = 0.91, p < 0.001), while the other treatments showed similar values over time, ranging from a minimum average of -17.89 ± 1.81 ‰ to a maximum average of -11.55 ± 5.32 ‰ $\delta$13C (Fig. 6A-D).

11) RC: Lines 20-21: "CO2 fluxes were also 10-fold higher in vegetated compared to adjacent, but bare sediments, indicating elevated microbial remineralization rates in vegetated sediments." Rewrite as words are repetitive and a bit confusing too. Given this statement minimal microbial description of these sediments will be very helpful.

AR: We thank the reviewer for the comment, we changed the sentence for clarification. Since we are unable to relate specific metabolic processes to specific microbial taxa, we have removed the term "microbial", and just refer to remineralization, as we cannot exclude contributions from other components of the benthic community.

AC: Page 9, line 15-17: Both CO2 and CH4 fluxes were higher in vegetated compared to adjacent bare sediments, indicating elevated remineralization rates in vegetated sediments as well as a higher susceptibility of seagrass sediment to increasing temperatures.

12) RC: Lines 34-35: "Mean CH4 fluxes at in situ temperature (25 ËŽC) in vegetated sediments were lower than the mean value of 85.09 ± 27.80 35 $\mu$mol CH4 m-2 d-1" - Caution needs to be exercised when expressing flux values to the second decimal. This is unnecessary given the uncertainties associated with flux estimates in general and large mean deviation in this particular case. ***

AR: We thank the reviewer for pointing this out, we have changed the sentence as follows:

AC: Page 9, line 28-29: Mean CH4 fluxes at in situ temperature (25 ËŽC) in vegetated sediments were lower than the mean value of 85.1 ± 27.8 $\mu$mol CH4 m-2 d-1 reported for other seagrass meadows in the Red Sea (Garcias-Bonet and Duarte, 2017).

[Figure]

---

## Author Comment (AC2) · 28 Oct 2019

Authors' response

RC: Comments from referees/public, AR: Authors' response, AC: Authors' changes in manuscript

ANONYMOUS REFEREE #2

Review report of the paper entitled "Warming enhances carbon dioxide and methane fluxes from Red Sea seagrass (Halophila stipulacea) sediments" submitted by Celina Burkholz et al to Biogeosciences

1) RC: Seagrass meadows, saltmarshes and mangroves are the "hot spot" blue carbon sinks and highly efficient in long-term carbon storage in the coastal marine ecosystems. Deterioration of these natural marine ecosystems through anthropogenic perturbation could change their carbon sinks efficiency and may contribute to climate change through re-emissions of locked carbon dioxide and other greenhouse gases. The warming-dependent emissions of metabolic GHG in the coastal ecosystems are likely to vary for coastal sediment of different geological origin. It is expected that in situ temperature increase is particularly important in seagrass meadows which have high carbon sequestration with long-term storage capacity and trap organic matter from external sources. The extreme conditions in the Red Sea i.e. slower seagrass growth due to nutrient limitation and greater microbial degradation of soil organic carbon because of high temperature could be related to low Corg storage in the sediment compared to temperate meadows. This paper reports the results of the study on the response of sediment collected from two H.stipulacea meadows, S1 (22ËŽ56.775'N, 38ËŽ52.677'E) and S2 (22ËŽ54.742'N, 38ËŽ53.848'E) at Al Kharar, a lagoon on the Saudi coast of the central Red Sea in February-March 2018, in terms of air-seawater fluxes of $CO_2$ and $CH_4$ along with their isotopic signature from Red Sea meadows compared to for gradual increase of temperature from 25 ËŽC to 37 ËŽC and prolonged darkness. However, I had difficulties to understand certain sections of the manuscript including the methodology and discussion. Furthermore, the author did not address appropriately other important elements particularly the redox conditions, anthropogenic pressure in and around the the ecosystems (Seagrass and bare soil).

AR: We thank the reviewer for the thorough review and constructive comments. We have addressed the comments individually to revise and clarify any uncertainties.

2) RC: The sediment in the present study is of carbonate (82.61-91.75%) dominated. What constitutes the rest fraction (9-18%) of the 1It does not represent all types of sediment of different geological origin in Al Kharar lagoon. Previous reports (Serrano et

al. 2018 Scientific Reports, 8:15037) indicate that soils in seagrass meadows in Saudi Arabia, Central Red Sea. are mainly constituted of clay and silt particles ($37 \pm 0.7\%$ on average), with a relatively high abundance of very fine sands ($21 \pm 0.4\%$) compared to fine sands ($16 \pm 0.4\%$), medium sands ($12 \pm 0.3\%$) and coarse sands ($14 \pm 0.7\%$). Youssef & El-Sorogy, 2016 ( Arab J Geosci 9:474) showed that sediment textures consist of mud, gravelly sand, and sandy mud and organic matter in the sediment may also be derived from Mangroves which are common in Al-Kharrar lagoon between latitude 22âŮę 45 and 23âŮę 00 N and longitude 39âŮę 00 and 38âŮę 45 E.

AR: We agree with the reviewer that this study does not represent all types of different geological origin in Al Kharar. Our main focus was to see a difference between two different sites, and then focus on the difference between vegetated and bare cores. We have therefore made the decision not to include other sediment types. We have added some additional information to the discussion section regarding the origin of organic matter content.

AC: Page 10, line 4-13: The isotopic signature of the CO2 released from bare sediments shifted with warming indicating a shift from seston (average $\delta$13C value of -25.43 $\pm$ 0.42 ‰ $Duarte et al., 2018) as the organic matter supporting respiration to seagrass carbon (average \delta$13C value of -7.73 $\pm$ 0.11 ‰ for Red Sea seagrass and -7.57 $\pm$ 0.15 ‰ for H. stipulacea in the Red Sea; Duarte et al., 2018) as the source of CO2. The mean $\delta$13C value of Red Sea seagrass sediments was reported to be $-13.36 \pm 0.4$ (Garcias-Bonet et al., 2019a), similar to the results found in this study. In the vegetated cores, the isotopic composition of CO2 stayed rather constant, indicating seagrass to be the main organic carbon source regardless of warming. Similar results were found in a recent study applying stable isotope mixing models, with the major contributors to the organic matter in seagrass sediments in the Red Sea being seagrass leaves and macroalgae blades, with contributions of 43 and 37 %, respectively (Garcias-Bonet et al., 2019a). We also observed a shift to a lighter isotopic signature of CH4 with warming, thereby

indicating an increasing CH4 production by methanogens with warming (Whiticar, 1990).

3) RC: Why you have collected sediment from two H.stipulacea meadows only?

AR: We thank the reviewer for the comment. Only two meadows were chosen for this hypothesis as the main focus was to test the effect of warming. Al Kharar is an enclosed lagoon where H. stipulacea forms dense patches and can be found growing along an OM gradient. We have chosen these two meadows to test our hypothesis that there is a difference along an OM gradient. Both meadows are monospecific, while many other meadows in Al Kharar are comprised of different seagrass species.

4) RC: Why the sediment of two H.stipulacea meadows showed high bulk density (1.1-1.28 g cm-3 ) relative to silt-clay sediment (1.05 g cm-3 ) in seagrass meadows in Saudi Arabia, Central Red Sea Serrano et al. 2018 observed. The bulk density of carbonate sediment (carbonate 0.7 g/cm3) should be even lower than that of sediment containing mainly sand (1.586 g cm-3) and claysilt ( 1.0 to 1.6 g/cm3). Organic carbon content of both vegetated (0.43 -0.55%) and bare (0.41 - 0.52%) carbonate sediment of H. stipulacea meadows is greater than the value reported for clay-silty sediment of other seagrass meadows (0.33%, Serrano et al. 2018 ).

AR: We thank the reviewer for pointing this out. Unfortunately, we don't have more data on sediment composition and grain size to confirm the other parts of the sediment composition. We acknowledge that the sediment composition differed from that reported in Serrano et al. (which includes Professor Carlos M. Duarte, under whose supervision both that and this research was conducted). We do not expect the sediments we sampled to be representative of all sediment configurations found in seagrass meadows in the Red Sea, nor Serrano et al. (2018) designed their research to encompass all sediment types occurring across the Red Sea.

5) RC: Why the production of H. stipulacea is high at the study site S2 compared to S1 and other seagrass meadows along the Saudi coast even though there was almost no

available nutrients (N & P), and no difference in nutrients between S1 and S2.

AR: We thank the reviewer for their concern. S2 had a higher organic matter content and a higher seagrass biomass compared to S1, which could have affected the fluxes. Additionally, previous studies have generally found a high variability in fluxes. We have rearranged this section to improve clarity.

AC: Page 8, line 20-page 10, line: 10: 4.1 Carbon dioxide and methane air-seawater fluxes The values reported for $CO_2$ and $CH_4$ fluxes varied greatly between the two sites studied here, with higher fluxes in the more organic sediments with higher biomass (S2). $CO_2$ and $CH_4$ fluxes were also highly variable over time in the studied site, as the first evaluation of fluxes in the same location delivered rates up to 100-fold above the rates of the second measurement one week later. Hence, organic matter availability along with temperature may account for the large variation in $CO_2$ and $CH_4$ fluxes. Additionally, the variability of $CO_2$ and $CH_4$ fluxes could also be supported by infaunal species present in the cores that were not recorded in this study. These trends were similar to results reported in previous studies, as a high variability between species and locations was found (cf. Table 1 in Garcias-Bonet and Duarte (2017)). Even though there were some differences, carbon, nitrogen and phosphorus concentrations were generally similar, and they didn't seem to have an effect on $CO_2$ and $CH_4$ fluxes. Carbon, nitrogen and phosphorus concentrations were low compared to mean values (Carbon: 33.6 $\pm$ 0.31 % DW, nitrogen: 1.92 $\pm$ 0.05 % DW, phosphorus: 0.23 $\pm$ 0.011 % DW) reported for seagrass leaves by Duarte (1990). Serrano et al. (2018) explained the discrepancy between Red Sea data and global data with the extreme conditions in the Red Sea, such as low nutrient input and high temperatures, as well as a limited data set favoring high carbon stocks in the Mediterranean. The results presented here add to those by Garcias-Bonet and Duarte (2017) to identify Red Sea seagrass communities as a significant source of $CH_4$. The presence of seagrass resulted in a higher organic matter supply to the sediments, favoring the presence of methanogens, which led to higher $CH_4$ fluxes compared to those fluxes supported in bare sediments (Barber

and Carlson, 1993; Bahlmann et al., 2015), consistent with the up to 100-fold higher $CH_4$ fluxes supported by vegetated compared to bare sediments in this study. Additionally, higher fluxes in vegetated cores could be an indicator of direct effects resulting from the presence of seagrass, as vascular plants on land have shown to have varying effects on methane emissions caused by differences in biomass and gross photosynthesis (Öquist and Svensson, 2002). Similar trends were also seen by Garcias-Bonet and Duarte (2017) who reported an increase in $CH_4$ fluxes with increasing organic matter content in Red Sea seagrass sediments. They reported organic matter contents in Red Sea seagrass sediments ranging from $2.33 \pm 0.07$ % (Halodule uninervis) to $12.42 \pm 0.23$ % (Enhalus acoroides), including a mixed meadow with H. stipulacea and H. uninervis showing a slightly higher organic matter content of $3.51 \pm 0.17$ % compared to vegetated sediments at S2. Moreover, they found the highest $CH_4$ fluxes in meadows with the highest biomass, confirming our findings with higher fluxes in study site S2. In terms of $CO_2$ equivalent greenhouse potential, only the bare sediment maintained at 25 ËŽC seemed to act as a C sink over the experimental period, while the vegetated sediments, both maintained at 25 ËŽC and exposed to warming, acted as sources of greenhouse gases. A sublethal disturbance, such as warming below the lethal threshold, can therefore lead to a shift of seagrass ecosystems from acting as net sinks to net sources of greenhouse gases, as demonstrated experimentally here.

6) RC: If seagrass detritus is the significant source of soil organic carbon, what is the source of organic carbon for bare sediment? Why vegetated and bare sediment of S1 showed no difference in soil organic carbon content?

AR: We thank the reviewer for their concern. Al Kharar is an enclosed lagoon receiving organic matter inputs from mangroves, macroalgae, phytoplankton, as well as from land occasionally. We have added some additional information to the discussion section

AC: Page 10, line 4-13: The isotopic signature of the $CO_2$ released from bare sediments shifted with warming indicating a shift from seston (average $\delta$13C value of -25.43 $\pm$ 0.42 ‰ $Duarte\,et\,al.,2018)$ $as\,the\,organic\,matter\,supporting\,respiration\,to\,seagrass\,carbon\,(average\,\delta$13C value of -7.73 $\pm$ 0.11 ‰ for Red Sea seagrass and -7.57 $\pm$ 0.15 ‰ for H. stipulacea in the Red Sea; Duarte et al., 2018) as the source of CO2. The mean $\delta$13C value of Red Sea seagrass sediments was reported to be $-13.36 \pm 0.4$ (Garcias-Bonet et al., 2019a), similar to the results found in this study. In the vegetated cores, the isotopic composition of CO2 stayed rather constant, indicating seagrass to be the main organic carbon source regardless of warming. Similar results were found in a recent study applying stable isotope mixing models, with the major contributors to the organic matter in seagrass sediments in the Red Sea being seagrass leaves and macroalgae blades, with contributions of 43 and 37 %, respectively (Garcias-Bonet et al., 2019a). We also observed a shift to a lighter isotopic signature of CH4 with warming, thereby indicating an increasing CH4 production by methanogens with warming (Whiticar, 1990).

7) RC: The $\delta$ 13 C-Corg showed depletion of 13C in comparison to leaf and both vegetated and bare sediment showed no significant difference of $\delta$ 13 C-Corg. Moreover, given $\delta$ 13C value of -7.96 $\pm$ 0.27 0/00 is lower than that of literature value for C4 plants -14 0/00. This needs clarifications. Use stable isotope mixing model to determine the actual contribution of seagrass on organic carbon content of the meadow sediment. CO2 fluxes were also 10-fold higher in vegetated compared to adjacent bare sediments, indicating elevated microbial remineralization rates in vegetated sediments.

AR: We share the reviewer's concern regarding the contributors to the organic matter in the sediment. Unfortunately, we cannot perform such analysis for the study site as we have not sampled all possible contributors for d13C analysis as it was out of the scope of this study. However, we are now discussing our results on the light of a comprehensive assessment of sources of organic carbon to sediments in Red Sea seagrass meadows we published since (Garcias-Bonet et al. (2019).

AC: Page 10, line 4-13: The isotopic signature of the CO2

released from bare sediments shifted with warming indicating a shift from seston (average $\delta$13C value of -25.43 $\pm$ 0.42 ‰ $Duarte\ et\ al., 2018)$ as the organic matter supporting respiration to seagrass carbon (average $\delta$13C value of -7.73 $\pm$ 0.11 ‰ for Red Sea seagrass and -7.57 $\pm$ 0.15 ‰ for H. stipulacea in the Red Sea; Duarte et al., 2018) as the source of CO2. The mean $\delta$13C value of Red Sea seagrass sediments was reported to be $-13.36 \pm 0.4$ (Garcias-Bonet et al., 2019a), similar to the results found in this study. In the vegetated cores, the isotopic composition of CO2 stayed rather constant, indicating seagrass to be the main organic carbon source regardless of warming. Similar results were found in a recent study applying stable isotope mixing models, with the major contributors to the organic matter in seagrass sediments in the Red Sea being seagrass leaves and macroalgae blades, with contributions of 43 and 37 %, respectively (Garcias-Bonet et al., 2019a). We also observed a shift to a lighter isotopic signature of CH4 with warming, thereby indicating an increasing CH4 production by methanogens with warming (Whiticar, 1990).

8) RC: P 4: Before measuring fluxes, why the water inside the cores was replaced by fresh seawater ? Is it accumulated pore water ? Replacement of pore water saturated with CO2 and CH4 by fresh sea water may result serious error!

AR: We thank the reviewer for this comment as we realized the text was not clear enough. We did not replace the pore water, we replaced the water overlying the sediment inside the cylindrical core, in order to be sure that measurements started with the same initial concentrations. We have changed the text to avoid confusion.

AC: Page 4, line 19-22: Before measuring fluxes, the water overlying the sediment inside the cores was carefully siphoned until only 5 mm of water remained over the sediment surface and fresh seawater was carefully siphoned in the core, to avoid disturbing the redoxcline, leaving a headspace of approx. 5 - 6 cm, and the cores were closed again with stoppers containing gas tight valves.
9) RC: Sediment core was incubated at one particular temperature and CO2 and CH4 concentrations were measured in 10 ml of head space air samples from each core at 0, 12hrs (light) and 12 hrs (dark). Considering, efluxes of 10,422 $\mu$mol CO2 m-2 d-1 and 88.11 $\mu$mol CH4 m-2 d-1, simple calculation shows after 12 hrs increased amount of CO2 and CH4 should be 37 $\mu$mol and 0.31 $\mu$mol, respectively. But standards (A: 750 ppm CO2, 9.7 ppm CH4, B: 250.5 ppm CO2, 3.25 ppm CH4) used before each (?) measurement of samples are all above those values. A plot of observed mixing ratio of CO2 and CH4 versus temperature should be given for better understanding.

AR: We thank the reviewer for their concern. We reported rates in this study, meaning that we calculated the difference between the first measurement (T0) and the 3rd measurement after 24 hours (T2). The actual measurements of the headspace air sample in ppm are not reported in this study. To avoid confusion, we have deleted the sentences referring to the light and dark fluxes, as we only report the daily fluxes. We changed the method section accordingly:

AC: Page 4, line 27-28: The daily CO2 and CH4 fluxes were calculated from the difference between T0 and T2 taking into account the core surface area ($\mu$mol m-2 d-1).

Page 5, line 34-page 6, line 2: The daily CO2 fluxes were calculated from the difference between T0 and T2 taking into account the core surface area ($\mu$mol m-2 d-1). Daily CH4 fluxes were estimated using the same calculations as for the CO2 fluxes with the exception of the Bunsen solubility coefficient.

10) RC: The mean ratio of CH4/CO2 was found to be âĹij0.008 which seems to be higher than the previous value reported for seagrass meadows (Halodule uninervis, Halodule pinifolia, Halophila ovalis, Halophila ovata, and Halophila beccarii ), Chilika Lagoon ( PLoS ONE 13(10): e0203922. https://doi.org/10.1371/journal.pone.0203922).

AR: We thank the reviewer for pointing this out. We've added additional information to the result and discussion section.

AC: Page 7, line 31: The CH4/CO2 ratio declined in the vegetated sediments exposed to warming from 7 to 0.8 %. Page 10, line 1-3: Increasing water temperature led to a decrease in the CH4/CO2 ratio. While there was ∼7 % of sequestered carbon released as CH4 to the atmosphere in vegetated sediments at 25 ËŽC (on day 2), it decreased to ∼0.8 % in vegetated sediments at 37 ËŽC. In contrast, Banerjee et al. (2018) reported ∼1% of carbon being released as CH4.

11) RC: The organic matter content was higher in S2 (vegetated 0.55%) than in S2 (bare 0.52%) by 5% . Is it below the error limit of its determination?

AR: We thank the reviewer for the comment. We found slightly higher organic matter content in S2 compared to S, in both vegetated and bare sediments, and a t-test was used to determine significance. We have changed this section accordingly.

AC: Page 6, line 32-33: The organic matter content was slightly higher in S2 than in S1, in both vegetated (t-test, $p < 0.0001$) and bare (t-test, $p < 0.001$) sediments (Table 1).

12) RC: How do you explain 6-fold CO2 and 100-fold CH4 greater emission in S2 (vegetated) than in S2 (bare) ? Provide a plot of observed CH4 concentration versus temperature. AR: We thank the reviewer for pointing this out. A 6-fold higher CO2 flux was reported for vegetated compared to bare sediments in both S1 and S2 indicating the difference between vegetated and bare cores. The same trend was seen for CH4 fluxes with up to 100-fold higher fluxes in vegetated compared to bare sediments. A plot of observed CO2 and CH4 concentrations vs temperature is shown in Fig. 2. The second x-axis indicates the experimental temperature for the community exposed to warming from 25 - 37 ËŽC. We added some additional information to the discussion section. AC: Page 8, line 21-40: The values reported for CO2 and CH4 fluxes varied greatly between the two sites studied here, with higher fluxes in the more organic sediments with higher biomass (S2). CO2 and CH4 fluxes were also highly variable over time in the studied site, as the first evaluation of fluxes in the same location delivered rates

up to 100-fold above the rates of the second measurement one week later. Hence, organic matter availability along with temperature may account for the large variation in $CO_2$ and $CH_4$ fluxes. Additionally, the variability of $CO_2$ and $CH_4$ fluxes could also be supported by infaunal species present in the cores that were not recorded in this study. These trends were similar to results reported in previous studies, as a high variability between species and locations was found (cf. Table 1 in Garcias-Bonet and Duarte (2017)). Even though there were some differences, carbon, nitrogen and phosphorus concentrations were generally similar, and they didn't seem to have an effect on $CO_2$ and $CH_4$ fluxes. Carbon, nitrogen and phosphorus concentrations were low compared to mean values (Carbon: 33.6 $\pm$ 0.31 % DW, nitrogen: 1.92 $\pm$ 0.05 % DW, phosphorus: 0.23 $\pm$ 0.011 % DW) reported for seagrass leaves by Duarte (1990). Serrano et al. (2018) explains the discrepancy between Red Sea data and global data with the extreme conditions in the Red Sea, such as low nutrient input and high temperatures, as well as a limited data set favoring high carbon stocks in the Mediterranean. The results presented here add to those by Garcias-Bonet and Duarte (2017) to identify Red Sea seagrass communities as a significant source of $CH_4$. The presence of seagrass resulted in a higher organic matter supply to the sediments, favoring the presence of methanogens, which led to higher $CH_4$ fluxes compared to those fluxes supported in bare sediments (Barber and Carlson, 1993; Bahlmann et al., 2015), consistent with the up to 100-fold higher $CH_4$ fluxes supported by vegetated compared to bare sediments in this study. Additionally, higher fluxes in vegetated cores could be an indicator of direct effects resulting from the presence of seagrass, as vascular plants on land have shown to have varying effects on methane emissions caused by differences in biomass and gross photosynthesis (Öquist and Svensson, 2002).

13) RC: The Fig 2 D and E shows large scatter of data and drawing those straight lines have no use.

AR: We thank the reviewer for their comment. Fig 2 D and E show a decline in $CH_4$ fluxes over time when the sediments were maintained at 25 ËŽC, both in vegetated

(R2 = 0.43, p < 0.001, Fig. 2D) and, less strongly, bare sediments (R2 = 0.24, p < 0.01; Fig. 2E, Table S2). Lines represent a fitted linear model. We have added the following sentence to the figure heading for clarification:

AC: Page 17, line 7: Lines represent a fitted linear regression equation.

14) RC: The CH4 fluxes declined over time when the sediments were maintained at 25 ËŽC, both in vegetated ( Fig. 2D) and, less strongly, bare sediments. In contrast, CH4 fluxes tended to increase with temperature in vegetated (Fig. 2D) and bare (Fig. 2E) sediments gradually warmed from 25 ËŽC to 37 ËŽC, although it was not significant. Since the in situ redox condition of both water and sediment was not maintained during experiment that may affect the equilibrium between counteracting microbial processes of production and oxidation in the sediment. The study on the response of benthic net methane concentrations to higher temperatures needs also to take into account methane production rates, Q10 values, and community sizes of methanogens and methanotrophs in seagrass sediments. I believe that the manuscript needs significant revision before being considered for resubmission.

AR: We thank the reviewer for the thorough review and constructive comments. We have addressed the comments individually, and we have revised the discussion section accordingly.

Specific comments

Abstract:

15) RC: In the first sentence please mention if Seagrasses are net source of CO2 and CH4.

AR: We thank the reviewer for their comment, the first sentence mentions that seagrasses can be both, sources and sinks of CO2 and CH4. We have changed the sentence for better understanding.

AC: Page 1, line 9-10: Seagrass meadows are autotrophic ecosystems acting as car-

bon sinks, but they have also been shown to be sources of carbon dioxide ($CO_2$) and methane ($CH_4$).

16) RC: Line 17: "We detected distinct differences between vegetated and unvegetated sediments, with the vegetated sediments supporting 6-fold higher $CO_2$ fluxes, and 10- to 100-fold higher $CH_4$ fluxes" This is a confusing statement. What are the conditions for the high flux was not mentioned?

AR: We thank the reviewer for pointing this out. The sentence only relates to the difference between vegetated and bare sediment, not different conditions. We changed the sentence for clarification.

AC: Page 1, line 17-18: We detected 6-fold higher $CO_2$ fluxes in vegetated compared to bare sediments, as well as 10- to 100-fold higher $CH_4$ fluxes.

Methodology

17) RC: "Assessment of carbon dioxide and methane air-seawater fluxes" This section is not clear. Please clarify the total number of samples collected from each core.

AR: We thank the reviewer for their comment. Three samples were taken from each core. The following sentence was changed for clarification: AC: Page 4, line 26-27: One sample from each core was taken at the start (T0), after 12 hours of light (T1) and after 12 hours of dark (T2). 18) RC: Was there a periodic collection?

AR: When referring to the collection of cores, cores were collected in February (comparison S1 and S2), March (temperature) and May (darkness) 2018. Samples from each core were collected taken at the start (T0), after 12 hours of light (T1) and after 12 hours of dark (T2). After the 24 hours measuring period, cores had time to acclimate to the new temperature before another measurement period happened.

19) RC: Did the cores have seagrass in the top surface? What was the depth of the core sample?

AR: We thank the reviewer for pointing out the missing information. Yes, there was seagrass in the vegetated sediments. We added the depth to the text for clarification.

AC: Page 3, line 18-20: Two H.stipulacea meadows at a depth of 2-3 m, S1 (22 ÌŁ56.775'N, 38 ÌŁ52.677'E) and S2 (22 ÌŁ54.742'N, 38 ÌŁ53.848'E), were chosen to represent a range 20 of organic matter content in the sediment, selected to evaluate greenhouse gas fluxes.

20) RC: What were the criteria for fixing the light condition at 70 $\mu$mol photons m-2 s-1 and 200 $\mu$mol photons m-2 s-1 at different incubation conditions?

AR: 70 $\mu$mol photons m-2 s-1 was the setting in the incubator chambers where the samples stayed only during the measurements, while the cores were exposed to 200 $\mu$mol photons m-2 s-1 in the aquaria between sampling days.

21) RC: "The temperature in the second aquarium was increased at a rate of 1 ËŽC day-1." Why this was done? Is this a natural increase (with 1day time) for the physiological adaptation by of seagrass?

AR: We thank the reviewer for pointing out the missing information. A temperature increase of 1 ËŽC d-1 was chosen to allow the seagrass to adjust to the higher temperature instead of creating stress by raising the temperature abruptly. We have added the missing information.

AC: Page 4, line 37-38: The temperature in the second aquarium was increased at a rate of 1 ËŽC day-1 to allow for acclimatization of the vegetated and bare cores.

22) RC: I wander how they have measured water-air flux from a system enclosed with sediment- water-air.

AR: We thank the reviewer for his/her comment. We followed the headspace technique as described in Garcias-Bonet et al. (2017) and Sea et al. (2018) that allows to measure air-sea fluxes: The water inside the cores overlying the sediment was replaced by fresh seawater leaving a headspace, and the cores were closed again with stoppers

containing gas tight valves. The cores were then left for one hour to allow for equilibration between the seawater and the headspace air. 10 mL of air were sampled from each core using a syringe, and injected the air sample in a cavity ring-down spectrometer through a small sample isotopic module extension, which provided both the partial pressure and the isotopic carbon composition of the $CO_2$ and $CH_4$ in the air sample.

23) RC: Lots of Q10 values are available in the literature. Do those values agree with the present report?

AR: We thank the reviewer for his/her comment. Q10 values in the literature indicate that respiration rates in seagrass have a higher temperature dependence compared to photosynthesis, while methanogenesis has a higher thermal dependence compared to photosynthesis and respiration. Our results agree with these findings as we saw an increase in $CO_2$ and $CH_4$ fluxes with warming. We added some additional information to the result and the discussion section:

AC: Page 7, line 31-33: For $CO_2$ and $CH_4$ fluxes in vegetated sediments, the Q10 value for the temperature range 25-37 ËŽC was 9 and 1.5, respectively, while the Q10 value for bare sediments was 13.8 and 4.2, respectively.

Page 9, line 32-33: Additionally, previous research has shown that methanogenesis has a higher thermal dependence than respiration and photosynthesis (Yvon-Durocher et al., 2014) confirming the trends seen here with increasing fluxes at higher temperatures.

Results:

24) RC: The first sentence needs to be revised.

AR: We thank the reviewer for pointing this out. We have changed the sentence for clarification.

AC: Page 6, line 26-27: Carbon, nitrogen (N), and phosphorus (P) concentrations in seagrass leaves were low, but they were 4- to 40-fold higher than vegetated and bare

sediment concentrations (Table 1).

25) RC: Page 6 line 28: "There were no consistent differences in C, N and concentration in bare and vegetated sediments (Table 1)." Does it indicate limited influence of additional carbon storage in the seagrass sediment? "The mean C, N content is significantly lower in the seagrass leaf (Table 1) compared to global average carbon content of 35% (Duarte 1990; Fourqurean et al. 2012).

AR: We thank the reviewer for his/her concern. Serrano et al. (2018) also reported low organic carbon content for Red Sea seagrass sediments due to the extreme conditions in the Red Sea (high temperatures and low nutrient input).

AC: Page 8, line 29-34: Even though there were some differences, carbon, nitrogen and phosphorus concentrations were generally similar, and they didn't seem to have an effect on $CO_2$ and $CH_4$ fluxes. Carbon, nitrogen and phosphorus concentrations were low compared to mean values (Carbon: 33.6 $\pm$ 0.31 % DW, nitrogen: 1.92 $\pm$ 0.05 % DW, phosphorus: 0.23 $\pm$ 0.011 % DW) reported for seagrass leaves by Duarte (1990). Serrano et al. (2018) explains the discrepancy between Red Sea data and global data with the extreme conditions in the Red Sea, such as low nutrient input and high temperatures, as well as a limited data set favoring high carbon stocks in the Mediterranean.

26) RC: There are several other seagrass sps apart from Halophila sp. in Kharar lagoon with considerable spatial variation sediment composition. Those Sps should also directly or indirectly affect the sediment composition. The sediment in the present study is manly composed of coral sand. Is it possible for the occurrence of such high concentration of OM in the sediment with no clay and silt?

AR: Halophila sp. is by far the dominant species in Al Kharar lagoon, forming a large monospecific meadow. Mixed meadows do occur, but the contribution of species other than Halophila is anecdotal at the ecosystem level. We did not measure the grain size distribution of the sediments sampled, so we have no basis to suggest that there was

no clay or silt, as the reviewer infers (but we did not suggest such thing).

27) RC: Page 6 Line 38: "The daily CO2 flux was up to 6-fold higher in vegetated compared to bare sediments, and tended to be generally higher in S2 compared to S1, where bare sediments showed net CO2 uptake, although differences were not significant" This result need to be discussed in details in the discussion section.

AR: We thank the reviewer for pointing this out. We rearranged the discussion section accordingly.

AC: Page 8, line 20-page 9, line 10: 4.1 Carbon dioxide and methane air-seawater fluxes The values reported for CO2 and CH4 fluxes varied greatly between the two sites studied here, with higher fluxes in the more organic sediments with higher biomass (S2). CO2 and CH4 fluxes were also highly variable over time in the studied site, as the first evaluation of fluxes in the same location delivered rates up to 100-fold above the rates of the second measurement one week later. Hence, organic matter availability along with temperature may account for the large variation in CO2 and CH4 fluxes. Additionally, the variability of CO2 and CH4 fluxes could also be supported by infaunal species present in the cores that were not recorded in this study. These trends were similar to results reported in previous studies, as a high variability between species and locations was found (cf. Table 1 in Garcias-Bonet and Duarte (2017)). Even though there were some differences, carbon, nitrogen and phosphorus concentrations were generally similar, and they didn't seem to have an effect on CO2 and CH4 fluxes. Carbon, nitrogen and phosphorus concentrations were low compared to mean values (Carbon: 33.6 ± 0.31 % DW, nitrogen: 1.92 ± 0.05 % DW, phosphorus: 0.23 ± 0.011 % DW) reported for seagrass leaves by Duarte (1990). Serrano et al. (2018) explains the discrepancy between Red Sea data and global data with the extreme conditions in the Red Sea, such as low nutrient input and high temperatures, as well as a limited data set favoring high carbon stocks in the Mediterranean. The results presented here add to those by Garcias-Bonet and Duarte (2017) to identify Red Sea seagrass communities as a significant source of CH4. The presence of seagrass resulted in a higher organic

matter supply to the sediments, favoring the presence of methanogens, which led to higher CH4 fluxes compared to those fluxes supported in bare sediments (Barber and Carlson, 1993; Bahlmann et al., 2015), consistent with the up to 100-fold higher CH4 fluxes supported by vegetated compared to bare sediments in this study. Additionally, higher fluxes in vegetated cores could be an indicator of direct effects resulting from the presence of seagrass, as vascular plants on land have shown to have varying effects on methane emissions caused by differences in biomass and gross photosynthesis (Öquist and Svensson, 2002). Similar trends were also seen by Garcias-Bonet and Duarte (2017) who reported an increase in CH4 fluxes with increasing organic matter content in Red Sea seagrass sediments. They reported organic matter contents in Red Sea seagrass sediments ranging from 2.33 ± 0.07 % (Halodule uninervis) to 12.42 ± 0.23 % (Enhalus acoroides), including a mixed meadow with H. stipulacea and H. uninervis showing a slightly higher organic matter content of 3.51 ± 0.17 % compared to vegetated sediments at S2. Moreover, they found the highest CH4 fluxes in meadows with the highest biomass, confirming our findings with higher fluxes in study site S2. In terms of CO2 equivalent greenhouse potential, only the bare sediment maintained at 25 ËŽC seemed to act as a C sink over the experimental period, while the vegetated sediments, both maintained at 25 ËŽC and exposed to warming, acted as sources of greenhouse gases. A sublethal disturbance, such as warming below the lethal threshold, can therefore lead to a shift of seagrass ecosystems from acting as net sinks to net sources of greenhouse gases, as demonstrated experimentally here.

28) RC: Page 7 Line 21: "Despite CO2 and CH4 fluxes showing the same response to warming in both types of sediment, vegetated sediments held higher fluxes than bare sediments. The relationship between net CO2 and CH4 fluxes in bare vs. vegetated sediments showed that both bare and vegetated communities tended to act as net CO2 sinks at 25 ËŽC, but tended to act as CO2 sources at warmer temperatures (Fig. 3A), whereas net CH4 fluxes were 3- to 8-fold higher in vegetated compared to bare sediments. (Fig. 3B)." The organic carbon contents are comparable for seagrass and bare soil. Do the result indicate higher susceptibility of the seagrass soil carbon at a

higher temperature?

AR: We agree with the reviewer that this could be an indication for higher susceptibility of seagrass sediment to higher temperatures. We have added some additional information in the discussion section.

AC: Page 9, line 13-15: Both $CO_2$ and $CH_4$ fluxes were higher in vegetated compared to adjacent bare sediments, indicating elevated remineralization rates in vegetated sediments as well as a higher susceptibility of seagrass sediment to increasing temperatures.

29) RC: Apart from regulation by Seagrass, $CH_4$ and $CO_2$ efflux depend on the redoxcline which has not been highlighted in relation to the abundance of methanogens and methanotrops. In methodology of the incubation experiment it is mentioned that ". . ..the water inside the cores was replaced by fresh seawater from the aquaria leaving a headspace of approx. 5 - 6 cm. . ..". Did this replacement change the redox condition of surface water?

AR: We thank the reviewer for his/her concern, which is due to insufficient detail in our account of procedures. This replacement was done gently to avoid disturbing the redoxcline. The replacement of the water was done through carefully siphoning the water in the core, retaining 5 mm of water overlaying the corer to avoid disturbing the redoxcline. Fresh seawater was then also carefully siphoned into the corer. Changing the water overlying the sediment allowed us to start with similar values at the first sampling (T0) to then see how the $CO_2$ and $CH_4$ fluxes were affected over a 24 hours period. Since we used the air from the headspace to determine air-water fluxes, we did not highlight the redoxcline in the water column. The replacement with freshwater would not change the redoxcline relative to that present in the environment, whereas leaving the water that was trapped inside the corers closed and locked could lead to accumulation of products released from the sediment that do not accumulate in the natural environment, where water flows freely and air-sea exchange operates. Hence,

removing the water that was contained in closed corers and replacing it with fresh sea-water is a standard procedure in sediment core incubations (e.g. Foster and Fullweiler 2019, for a recent paper), necessary to avoid, rather than introduce, artifacts. We now specify how this replacement was done.

AC: Page 4, line: 19-22: Before measuring fluxes, the water overlying the sediment inside the cores was carefully siphoned until only 5 mm of water remained over the sediment surface and fresh seawater was carefully siphoned in the core, to avoid disturbing the redoxcline, leaving a headspace of approx. 5 - 6 cm, and the cores were closed again with stoppers containing gas tight valves.

Reference: Foster, S.Q. and Fulweiler, R.W., 2019. Estuarine Sediments Exhibit Dynamic and Variable Biogeochemical Responses to Hypoxia. Journal of Geophysical Research: Biogeosciences, 124(4), pp.737-758.

Discussion:

30) RC: This section is inadequate and the observed results need more detailed discussions about the variations $CO_2$ and $CH_4$ fluxes at natural, high temperature and low light conditions. "Similar trends were seen by Garcias-Bonet and Duarte (2017) who reported an increase in $CH_4$ fluxes with increasing organic matter content in Red Sea seagrass sediments." In the present study lower $CO_2$ and $CH_4$ fluxes were recorded from adjacent bare sediments with comparable Organic C content.

AR: We thank the reviewer for his/her comment. We have changed the discussion according to the comments. We based this statement on the fact that the vegetated sediment of S2 had a higher OM content than bare sediments of S2, as well as vegetated and bare sediments in S1. Concurrently, these were the cores with the highest fluxes.

AC: Page 8, line 21-page 9, line 10: The values reported for $CO_2$ and $CH_4$ fluxes varied greatly between the two sites studied here, with higher fluxes in the more organic

sediments with higher biomass (S2). $CO_2$ and $CH_4$ fluxes were also highly variable over time in the studied site, as the first evaluation of fluxes in the same location delivered rates up to 100-fold above the rates of the second measurement one week later. Hence, organic matter availability along with temperature may account for the large variation in $CO_2$ and $CH_4$ fluxes. Additionally, the variability of $CO_2$ and $CH_4$ fluxes could also be supported by infaunal species present in the cores that were not recorded in this study. These trends were similar to results reported in previous studies, as a high variability between species and locations was found (cf. Table 1 in Garcias-Bonet and Duarte (2017)). Even though there were some differences, carbon, nitrogen and phosphorus concentrations were generally similar, and they didn't seem to have an effect on $CO_2$ and $CH_4$ fluxes. Carbon, nitrogen and phosphorus concentrations were low compared to mean values (Carbon: $33.6 \pm 0.31$ % DW, nitrogen: $1.92 \pm 0.05$ % DW, phosphorus: $0.23 \pm 0.011$ % DW) reported for seagrass leaves by Duarte (1990). Serrano et al. (2018) explains the discrepancy between Red Sea data and global data with the extreme conditions in the Red Sea, such as low nutrient input and high temperatures, as well as a limited data set favoring high carbon stocks in the Mediterranean. The results presented here add to those by Garcias-Bonet and Duarte (2017) to identify Red Sea seagrass communities as a significant source of $CH_4$. The presence of seagrass resulted in a higher organic matter supply to the sediments, favoring the presence of methanogens, which led to higher $CH_4$ fluxes compared to those fluxes supported in bare sediments (Barber and Carlson, 1993; Bahlmann et al., 2015), consistent with the up to 100-fold higher $CH_4$ fluxes supported by vegetated compared to bare sediments in this study. Additionally, higher fluxes in vegetated cores could be an indicator of direct effects resulting from the presence of seagrass, as vascular plants on land have shown to have varying effects on methane emissions caused by differences in biomass and gross photosynthesis (Öquist and Svensson, 2002). Similar trends were also seen by Garcias-Bonet and Duarte (2017) who reported an increase in $CH_4$ fluxes with increasing organic matter content in Red Sea seagrass sediments. They reported organic matter contents in Red Sea seagrass sediments ranging from

2.33 ± 0.07 % (Halodule uninervis) to 12.42 ± 0.23 % (Enhalus acoroides), including a mixed meadow with H. stipulacea and H. uninervis showing a slightly higher organic matter content of 3.51 ± 0.17 % compared to vegetated sediments at S2. Moreover, they found the highest $CH_4$ fluxes in meadows with the highest biomass, confirming our findings with higher fluxes in study site S2.

Page 9, line 18-23: However, the fluxes maintained at 25 ĚŽC showed a net $CO_2$ uptake with a mean of 464.78 ± 156.6 $\mu$mol $CO_2$ m-2 d-1 (Table S1), while those reported in a mixed Halodule sp. and Halophila sp. meadow in India showed a net $CO_2$ release (dry season: 1,190 ± 1,600 $\mu$mol $CO_2$ m-2 d-1, wet season: 18,400 ± 8,800 $\mu$mol $CO_2$ m-2 d-1; Banerjee et al., 2018). Both values reported were measured at higher temperatures (dry season: 30 ± 0.68 ĚŽC, wet season: 27.94 ± 0.72 ĚŽC, Banerjee et al., 2018) compared to our fluxes measured at 25 ĚŽC, also indicating that temperature might lead to higher fluxes.

Page 9, line 26-31: Mean $CH_4$ fluxes at in situ temperature (25 ĚŽC) in vegetated sediments were lower than the mean value of 85.1 ± 27.8 $\mu$mol $CH_4$ m-2 d-1 reported for other seagrass meadows in the Red Sea (Garcias-Bonet and Duarte, 2017). In contrast, the community exposed to warming reached a maximum average $CH_4$ flux almost 4-fold higher than the community held at 25 ĚŽC, and showed a clear increase with warming, relative to sediments held at 25 ĚŽC. The increase in $CH_4$ fluxes with warming was consistent with reports from Barber and Carlson (1993) for a Thalassia testudinum community in Florida Bay and Garcias-Bonet and Duarte (2017) for Red Sea seagrass communities, who reported higher $CH_4$ production rates at higher temperatures.

31) RC: Page 8 Line 24: ". . ..while the $CO_2$ fluxes in vegetated sediments maintained at 25 ĚŽC decreased over time." Why it was high initially and decreased gradually? explain.

AR: We thank the reviewer for the comment. The initial higher value might have been

a response to the stress due to sample collection and transportation. Even though we allowed the cores time to adapt, this value could still be an indicator for the experienced disturbance. All values measured after more than 5 days of sampling showed negative values indicating the capacity of seagrass sediments to act as carbon sinks. We added the following information to the discussion section.

AC: Page 9, line 23-25: An initial high CO2 flux measured on day 2 after sampling could be an indicator for the experienced disturbance due to sample collection and transportation even though we allowed the cores some time to adapt.

32) RC: Line 30: "The presence of seagrass resulted in a higher organic matter supply to the sediments, favoring the presence of methanogens, which led to higher CH4 fluxes compared to those fluxes supported in bare sediments (Barber and Carlson, 1993; Bahlmann et al., 2015), consistent with the up to 100-fold higher CH4 fluxes supported by vegetated compared to bare sediments in this study." In the present study, soil organic C content in vegetated and adjacent bare sediment are comparable but CH4 fluxes are significantly different. Does the results (high emissions at S1 and S2) indicated direct CH4 emission from Seagrass? (pls see:Quist and Svensson, 2002, Vascular plants as regulators of methane emissions from a subarctic mire ecosystem, jgr, 107, NO. D21, 4580 and others) Effect of warming

AR: We thank the reviewer for the comment. We have rearranged the discussion section accordingly and added some additional information:

AC: Page 8, line 34-40: The results presented here add to those by Garcias-Bonet and Duarte (2017) to identify Red Sea seagrass communities as a significant source of CH4. The presence of seagrass resulted in a higher organic matter supply to the sediments, favoring the presence of methanogens, which led to higher CH4 fluxes compared to those fluxes supported in bare sediments (Barber and Carlson, 1993; Bahlmann et al., 2015), consistent with the up to 100-fold higher CH4 fluxes supported by vegetated compared to bare sediments in this study. Additionally, higher fluxes in

vegetated cores could be an indicator of direct effects resulting from the presence of seagrass, as vascular plants on land have shown to have varying effects on methane emissions caused by differences in biomass and gross photosynthesis (Öquist and Svensson, 2002).

33) RC: Page 8, Line 26 : "However, the fluxes maintained at 25 ËŽC were about 3-fold below those reported in a mixed Halodule sp. and Halophila sp. meadow in India (980 $\mu$mol CO2 m-2 d-1; Banerjee et al., 2018)." Is this the mean flux (the present value) or the least value recorded or the values represent during the end of the incubation at 25 ËŽC? In the previous sentence it was mentioned as "CO2 fluxes in vegetated sediments maintained at 25 ËŽC decreased over time"

AR: We thank the reviewer for pointing out this error. We have edited the sentence accordingly.

AC: Page 9, line 18-23: However, the fluxes maintained at 25 ËŽC showed a net CO2 uptake with a mean of 464.78 $\pm$ 156.6 $\mu$mol CO2 m-2 d-1 (Table S1), while those reported in a mixed Halodule sp. and Halophila sp. meadow in India showed a net CO2 release (dry season: 1,190 $\pm$ 1,600 $\mu$mol CO2 m-2 d-1, wet season: 18,400 $\pm$ 8,800 $\mu$mol CO2 m-2 d-1; Banerjee et al., 2018). Both values reported were measured at higher temperatures (dry season: 30 $\pm$ 0.68 ËŽC, wet season: 27.94 $\pm$ 0.72 ËŽC, Banerjee et al., 2018) compared to our fluxes measured at 25 ËŽC, also indicating that temperature might lead to higher fluxes.

34) RC: Page 8, Line 40: "We also reported a 10-fold decline in CH4 fluxes over time for sediment communities maintained at 25 ËŽC, which could be attributable to increased sulfate reduction, reduced CH4 production or a combination of both. Methane is produced under anoxic conditions in marine sediments, yet only a small portion is released, as CH4 production by methanogens is compensated for by CH4 consumption by sulfatereducing bacteria (Barnes and Goldberg, 1976)." The soil redox conditions were not measured in this study. Please show some evidence of enhanced soil anoxicity which may have increased SO4 reduction. What could be the other reason/s of reduced CH4 production.

AR: We thank the reviewer for the comment. We have added more information to this paragraph to improve clarity and address the reviewer's concern.

AC: Page 9, line 34-40: We also reported a 10-fold decline in CH4 fluxes over time for sediment communities maintained at 25 ËŽC, which could be attributable to increased sulfate reduction, reduced CH4 production or a combination of both. Methane is produced under anoxic conditions in marine sediments, yet only a small portion is released, as CH4 production by methanogens is compensated for by CH4 oxidation by sulfate-reducing bacteria (Barnes and Goldberg, 1976). Similar to the trends seen in CO2 fluxes, the decrease in CH4 fluxes could be attributable to an initial stress response to the disturbance caused by sample collection and transportation. While reduced photosynthetic activity and a decrease in biomass could result in higher CH4 fluxes (Lyimo et al., 2018), the cores maintained at 25 ËŽC might show the effect of healthy conditions.

Conclusion:

35) RC: This section may be rewritten after revising the discussion section with few more synthesizing statements.

AR: We thank the reviewer for the comment, we have changed the conclusion accordingly.

AC: Page 11, line 3-13: In summary, this study reports, for the first time, experimental evidence that warming leads to increased greenhouse gas (CO2 and CH4) fluxes in a H. stipulacea meadow in the Red Sea, and it may lead to seagrass meadows shifting from acting as sinks to sources of greenhouse gases. Increased fluxes at higher temperatures can be an indication of higher remineralization rates and a higher susceptibility of vegetated sediments to temperature. The elevated organic matter content, higher

biomass and higher plant activity in vegetated sediments led to increased CO2 and CH4 fluxes in vegetated compared to bare sediments and a much steeper increase in CO2 and CH4 fluxes with warming. In addition, prolonged darkness led to an increase in CO2 fluxes, while CH4 fluxes decreased over time, also indicating organic matter to be the driver. However, we also found a high variability in fluxes over time indicating that other factors, such as infaunal species, could play a role as well. While current focus is on conserving blue carbon ecosystems from losses due to deteriorated water quality or mechanical damage, our results show that sublethal warming may also lead to emissions of greenhouse gases from seagrass meadows, contributing to a feedback between ocean warming and further climate change.

36) RC: Figure 2: what does the blue dots represent?

AR: We thank the reviewer for pointing out the missing information, the blue dots represent "Constant temperature, 25 ËŽC". We have added the missing information to the graph on page 17.

37) RC: Figure 3: . . ... the dashed line indicates line 1:1, and dotted lines show lines 2:1, 4:1 and 8:1. Need to ve detailed.

AR: We thank the reviewer for pointing this out. We used the additional lines to visualize the relationship between vegetated and bare sediments showing that CH4 fluxes were 3- to 8-fold higher in vegetated compared to bare sediments.
* * *
[Figure]

CO$_2$

CH$_4$

A) Vegetated

D) Vegetated

B) Bare

E) Bare

C) Net fluxes

F) Net fluxes

— Constant temperature, 25 ˚C
— Warming, 25 - 37 ˚C
• Outlier

— Vegetated
— Bare

**Fig. 1.** Fig. 2